# A Comprehensive Review of Smart Thermosensitive Nanocarriers for Precision Cancer Therapy

**DOI:** 10.3390/ijms26157322

**Published:** 2025-07-29

**Authors:** Atena Yaramiri, Rand Abo Asalh, Majd Abo Asalh, Nour AlSawaftah, Waad H. Abuwatfa, Ghaleb A. Husseini

**Affiliations:** 1Biomedical Engineering Program, College of Engineering, American University of Sharjah, Sharjah P.O. Box 26666, United Arab Emirates; g00098847@aus.edu; 2Internal Medicine Residency, Internal Medicine Department, Manatee Memorial Hospital, Bradenton, FL 34208, USA; randabouassaleh@hotmail.com (R.A.A.); majdabouassaleh@hotmail.com (M.A.A.); 3Material Science and Engineering Ph. D. Program, College of Arts and Sciences, American University of Sharjah, Sharjah P.O. Box 26666, United Arab Emirates; g00051790@alumni.aus.edu (N.A.); g00062257@alumni.aus.edu (W.H.A.); 4Department of Chemical and Biological Engineering, College of Engineering, American University of Sharjah, Sharjah P.O. Box 26666, United Arab Emirates; 5Bioscience and Bioengineering Ph. D. Program, College of Engineering, American University of Sharjah, Sharjah P.O. Box 26666, United Arab Emirates

**Keywords:** thermosensitive nanocarriers, drug delivery, nanotechnology, hyperthermia, cancer treatment, smart drug delivery systems

## Abstract

By 2030, millions of new cancer cases will be diagnosed, as well as millions of cancer-related deaths. Traditional drug delivery methods have limitations, so developing smart drug delivery systems (SDDs) has emerged as a promising avenue for more effective and precise cancer treatment. Nanotechnology, particularly nanomedicine, provides innovative approaches to enhance drug delivery, including the use of nanoparticles. One such type of SDD is thermosensitive nanoparticles, which respond to internal and external stimuli, such as temperature changes, to release drugs precisely at tumor sites and minimize off-target effects. On the other hand, hyperthermia is a cancer treatment mode that goes back centuries and has become popular because it can target cancer cells while sparing healthy tissue. This paper presents a comprehensive review of smart thermosensitive nanoparticles for cancer treatment, with a primary focus on organic nanoparticles. The integration of hyperthermia with temperature-sensitive nanocarriers, such as micelles, hydrogels, dendrimers, liposomes, and solid lipid nanoparticles, offers a promising approach to improving the precision and efficacy of cancer therapy. By leveraging temperature as a controlled drug release mechanism, this review highlights the potential of these innovative systems to enhance treatment outcomes while minimizing adverse side effects.

## 1. Introduction

Among public health challenges, cancer poses a significant threat to individuals, families, healthcare systems, and society. It is estimated that millions of new cancer cases and deaths will occur globally by 2030. There is evidence that this trend is associated with sedentary lifestyles, poor diets, obesity, tobacco use, and air pollution [1]. Many cancer patients die because of tumor cell invasion of nearby tissues and distant organs (i.e., metastasis). There is no specific cause of cancer because specific processes trigger malignant transformation. However, numerous factors can contribute to disease, including radiation, chemicals, viruses, and inflammation [2].

Early diagnosis is cancer treatment’s first and most crucial phase. In most cases, surgery is the primary treatment option for solid tumors. Additionally, chemotherapy may eliminate residual cancer cells and potential metastases concurrently or after treatment [3]. In the traditional approach to cancer treatment, the entire body is treated without focusing on specific regions. This damages both cancerous cells and healthy cells. To prevent severe side effects, therapies are often administered at lower-than-ideal doses [4]. Therapeutic treatments are effective when active substances reach their precise targets at optimal concentrations with minimal side effects. Nevertheless, traditional methods of drug administration often fail because they rely on systemic circulation instead of considering the intricate dynamics within the target tissues. Controlled release systems are a promising approach for regulating drug access and release rates [5].

Traditional methods of drug delivery include oral ingestion and intravenous injection. Although effective, they have several drawbacks, including poor targeting, inconsistent release, high doses, fluctuating levels, and adverse side effects. To address these challenges, pharmaceutical researchers are developing more competitive drug delivery systems known as SDDs that are cost-effective, convenient, and compatible with the body’s natural processes [6]. Figure 1 shows the characteristics of drug delivery methods, including both conventional and targeted cancer therapies.

Nanotechnology is an emerging field that involves manipulating devices and structures at the atomic, molecular, and supramolecular levels. The application of nanotechnology in healthcare, known as nanomedicine, can revolutionize individual and population health by providing personalized approaches to disease screening, diagnosis, and treatment [7]. To develop anticancer nanomedicines, drugs must be incorporated into smart nanoparticles (NPs) capable of transporting and releasing the drugs at specific sites. Three significant steps are involved in developing a smart nanocarrier system: identifying cancer cells via active or passive targeting, releasing drugs from the nanocarriers in response to stimulation, and targeting cancer sites with engineered drug delivery systems or facilitating the controlled release of therapeutic agents [6]. Nanoparticles as drug carriers require sufficient biocompatibility, biodegradability, stability in physiological environments, and a high drug-loading capacity with minimal toxicity [8]. The different classes of smart nanoparticles include polymer-based carriers, lipid-based carriers, viral nanoparticles, carbon nanotubes, ceramic nanoparticles, and metal-based nanoparticles. Several polymer-based carriers, including polymeric nanoparticles, dendrimers, hydrogels, and micelles, have been explored for cancer treatment. Liposomes, solid lipid nanoparticles, and lipid nanoemulsions are lipid-based carriers that have been investigated for cancer therapy treatment [7].

SDDs can come in various sizes, including macroscopic, microscopic, and nanoscopic. By designing nanocarriers appropriately, drugs can be delivered systemically while selectively targeting specific cells. Nanocarriers carry drugs intravenously, modifying the distribution and clearance of drugs within the body by transporting them to specific tissues. While the drug is encapsulated in the nanocarrier, it cannot exit the bloodstream. Instead, it uses passive or active targeting mechanisms to access target tissues [5]. Drug delivery utilizes tumor biology characteristics, such as Enhanced Permeability and Retention (EPR), to passively deliver drugs to tumor cells. An active targeting strategy involves attaching a ligand to non-toxic drug carriers that target specific tumor markers, thereby enabling the delivery of the drug to the intended site of action. These strategies aim to increase drug delivery efficiency while minimizing off-target effects [9]. With smart nanoparticles, targeted drug delivery at tumor sites is enabled by the response to either internal (e.g., pH, enzymes, redox gradients) or external stimuli (e.g., light, temperature, ultrasound). Nanoparticles that respond to stimuli ensure precise drug release, enhancing therapeutic effects while minimizing off-target side effects. In intrinsic stimulus-responsive drug delivery systems (DDSs), drugs are loaded and released in response to tumor-specific factors, such as pH levels and enzyme activity. In contrast, extrinsic stimulus-sensitive DDSs respond to external triggers such as light, temperature, or magnetic fields to deliver targeted and controlled drug delivery at tumor sites. In response to external stimuli, DDSs undergo chemical or physical changes that allow precise control of tumor targeting, penetration, cellular uptake, and intracellular drug delivery. The extrinsic stimulus-responsive DDS may offer advantages such as precise control over drug release location and intensity. Still, they may not be suitable for treating distal or metastatic cancers with unknown tumor locations [10].

Temperature is one of the most extensively studied external physical stimuli for triggering drug release from smart nanocarriers. Thermosensitive nanoparticles (TSNPs) are specifically designed to respond to mild increases in temperature, typically in the range of 39–42 °C, which is just above the normal physiological temperature of 37 °C, by releasing their therapeutic payload. This method is particularly beneficial for treating solid tumors, as localized hyperthermia can be safely induced using various external energy sources, including magnetic hyperthermia (using alternating magnetic fields) [11], High-Intensity Focused Ultrasound [12], near-infrared (NIR) laser irradiation [13], and radiofrequency or microwave heating. By applying heat directly to the tumor site, TSNPs facilitate spatially and temporally controlled drug release. This targeted approach helps reduce off-target toxicity and enhances the effectiveness of anticancer treatments [14,15].

TSNPs are based on materials that undergo a physical or chemical transition in response to temperature. These transitions alter the nanoparticle’s structure or permeability, facilitating the release of encapsulated drugs. Polymer-based thermosensitive nanoparticles operate on the principle of reversible solubility transitions of thermoresponsive polymers in aqueous environments. This behavior is mainly controlled by two critical parameters: the Lower Critical Solution Temperature (LCST) and the Upper Critical Solution Temperature (UCST). Polymers exhibiting LCST behavior are soluble and hydrophilic below a certain temperature but become hydrophobic and aggregate when the temperature exceeds this threshold, due to the breakdown of hydrogen bonding and the predominance of hydrophobic interactions. A well-known example of this is poly(N-isopropylacrylamide) (PNIPAM), which has an LCST of approximately 32 °C. This temperature can be adjusted to between 40 and 42 °C through copolymerization, making PNIPAM suitable for controlled drug release applications. In these systems, a drug is retained within the swollen polymer matrix at normal body temperature (37 °C) and is released when the polymer shrinks above the LCST (around 42 °C). In contrast, UCST-type polymers are soluble at higher temperatures and become insoluble when the temperature drops, due to strong interchain interactions, such as hydrogen bonding or electrostatic interactions. Although UCST systems are less common in biomedical applications due to biocompatibility concerns, they can be useful for processes requiring inverse release patterns, such as drug release upon cooling. These polymers are utilized to design thermosensitive micelles, hydrogels, nanogels, and polymeric vesicles. For instance, micelles feature a hydrophobic core that destabilizes at elevated temperatures, facilitating the release of encapsulated drugs. Hydrogels, on the other hand, are three-dimensional polymer networks that either shrink or swell in response to temperature changes. Overall, polymer-based TSNPs provide a high degree of tunability, allowing for the loading of both hydrophilic and hydrophobic drugs. They can also be modified for biodegradability and targeted delivery, enhancing their potential for biomedical applications [16,17].

Lipid-based thermosensitive nanoparticles, particularly thermosensitive liposomes, are engineered using lipid bilayers that become more permeable when exposed to mild heat. These systems incorporate specific lipids with defined phase transition temperatures (Tm), such as DPPC (Tm ≈ 41.5 °C). Below this temperature, the lipid bilayer remains in a rigid, low-permeability gel phase. However, when the temperature rises slightly above this threshold—typically around 41–42 °C—the bilayer transitions into a more fluid, liquid–crystalline state. This change enhances membrane permeability and triggers rapid drug release. These liposomes take advantage of the EPR effect, allowing them to passively target tumors. By releasing their contents upon localized heating (for example, via magnetic resonance high-intensity focused ultrasound (MR-HIFU)), they enable precise, on-demand drug delivery directly at the tumor site [18,19].

This paper addresses the gap in understanding the synergistic effects of hyperthermia and thermosensitive nanocarriers, particularly in overcoming the challenges of targeted drug delivery while minimizing side effects. The subsequent sections of this paper are organized as follows: Section 2 covers temperature as a triggering mechanism, hyperthermia cancer therapy, and hyperthermia-induced drug release. Section 3 discusses thermosensitive nanoparticles, including polymer-based types such as thermosensitive micelles, hydrogels, and dendrimers, as well as lipid-based nanoparticles like thermosensitive liposomes and solid lipid nanoparticles. Section 4 explains the preclinical and clinical applications of thermosensitive nanoparticles for cancer therapy. Section 5 addresses the challenges in this field, specifically focusing on those encountered in clinical trials, and outlines future directions for addressing these challenges. Finally, Section 6 concludes the paper and highlights areas for future research. The novelty lies in integrating hyperthermia with various temperature-sensitive nanocarriers, such as micelles, hydrogels, dendrimers, liposomes, and solid lipid nanoparticles, with a focus on organic nanoparticles. The primary aim is to enhance the precision of cancer treatment through temperature-triggered drug release mechanisms.

## 2. Temperature as a Triggering Mechanism

The normal human body temperature ranges from 36.1 °C to 37.2 °C. Temperature increases outside of this range should be considered as a sign of fever, inflammation, infarctions, or tumors, all of which may cause a localized increase in temperature [5]. Compared to normal tissue, cancerous tissues exhibit abnormal temperature increases (from 37 °C to 39 °C), providing a potential intrinsic stimulus for drug delivery, which requires materials to be precise and adjustable [10]. Temperature-responsive DDSs are widely used in cancer treatment. Compared to other stimuli, temperature offers the most convenient and effective method of controlled drug release. Considering the temperature difference between cancerous and normal tissues, functionalized nanoparticles can be triggered to enhance their drug release [8]. At normal body temperatures (up to 37 °C), these systems remain stable and retain their drug cargoes, but release them at higher temperatures (e.g., >40 °C) due to physiological changes or external stimuli [10].

### 2.1. Hyperthermia Cancer Therapy

Hyperthermia is a Greek word composed of “hyper”, meaning to elevate, and “therme,” meaning heat. Going back to around 3000 B.C., the use of heat to cure cancer was performed through several approaches, like hot water, sand (mud baths), hot air, steam, and “fire drills” (hot blades and sticks) to burn cancerous cells [20]. The Greek philosopher Parmenides believed that fever could be used to treat various diseases, including cancer. Many researchers discovered that hyperthermia (the ability to elevate body temperature) significantly affects cancer treatment [21]. Hyperthermia has been defined as a phenomenon that increases body temperature locally or throughout the body from 40 °C to 43 °C [22], with the use of internal or external applications to treat various diseases, such as cancer [15].

One of the most significant advantages of hyperthermia is its ability to selectively kill cancerous cells while causing minimal damage to normal cells. Cancer cell structures are compact and disorganized to prevent heat dissipation. Increasing the temperature of cells above 42 °C increases their sensitivity to chemotherapy and radiotherapy. Among other cancer treatments, hyperthermia is used as a complementary therapy. As a result of hyperthermia treatment, tumor blood flow, vascular permeability, and vessel pore size are improved, allowing drugs to be delivered more efficiently to the tumor site. Treatment duration and temperature during therapeutic sessions determine the effectiveness of hyperthermia in killing cancer cells. To maximize treatment efficacy and avoid potential side effects, hyperthermia sessions should be carefully controlled and monitored. There are three types of hyperthermia treatment: localized, regional, and whole-body treatments that are applied to either localized, advanced, or widespread cancer. Local hyperthermia treats superficial or easily accessible tumors. The regional form treats advanced tumors that affect large areas of the body or organs [23]. Whole-body hyperthermia is a treatment for metastatic cancers that involves raising the body’s temperature to 42.0 °C to target and destroy cancer cells. One of the main challenges of this approach is efficiently delivering heat while minimizing energy loss. Modern techniques for generating therapeutic heat include ultrasound, infrared radiation, radiofrequency, and microwave electromagnetic energy, which can be used separately or in combination [24].

There are two different techniques for local hyperthermia: thermal ablation and mild hyperthermia. In thermal ablation, the temperature applied exceeds 50 °C, causing severe damage to the cellular structure and resulting in tissue destruction. This strategy can effectively eliminate tumor cells restricted to a specific organ. In contrast, mild hyperthermia uses a temperature of around 41 °C to 45 °C. The effects of mild hyperthermia on tumor cells are both direct and indirect. As a result of oxygen-deficient and acidic conditions inside tumors with poor blood flow, cells are more sensitive to heat. They are, therefore, more vulnerable to this type of treatment. Mild hyperthermia increases the permeability of cell membranes, disrupts DNA repair capabilities, and enhances tumor perfusion. By combining hyperthermia with chemotherapy and radiation, tumor cells are more susceptible to drug delivery and have increased sensitivity to chemotherapy and radiation [14].

Hyperthermic tumor therapy (HTT) induces changes in tumor tissues that enhance the effectiveness of conventional cancer treatments. By increasing membrane fluidity, tumor cells become more permeable to drugs, while disruptions in intracellular ion levels (such as Ca^2+^, Na^+^, K^+^, and Mg^2+^) trigger cell death pathways. HTT raises interstitial fluid pressure (IFP) and reduces blood flow in the tumor; however, it can also lower IFP by improving lymphatic drainage and blood perfusion, thereby enhancing nanoparticle (NPs) delivery. Additionally, HTT increases the production of reactive oxygen species (ROS), promoting drug uptake and apoptosis. Nanoparticle-assisted HTT results in greater tumor cell death due to collagen damage, with imaging studies indicating that heat expands tumor porosity for improved NPs distribution [25].

Several factors contribute to local hyperthermia, including the energy source, various energy transfer methods, the size of the heated volume, whether the energy is applied externally or internally, and the temperature control range. In order to provide better spatial control of heating, hyperthermia systems utilize more complex heating and temperature monitoring technology [26,27].

Several hyperthermia systems are applied to heat the cancerous tissue externally. These systems are based on electromagnetics, radiofrequency, or ultrasound. Hyperthermia systems based on electromagnetic energy are among the most prevalent ones to heat the tumor site and its surrounding tissue [28]. Radiative hyperthermia systems are considered advanced technology because they are capable of dynamically distributing spatial energy. Additionally, radiofrequency-based hyperthermia systems can be used to heat tumors both superficially and deeply [29]. In medical practice, HIFU is increasingly used for procedures requiring high temperatures, such as thermal ablation, but less frequently for mild hyperthermia treatment [30]. As a result of HIFU, pressure waves are applied to induce small tissue vibrations, which result in heat generation. HIFU has a number of advantages over electromagnetic (EM) heating, including its ability to penetrate deeply and provide precise focal heating [31]. However, HIFU cannot be used in certain areas, such as the lungs or rib cage, due to significant reflections at air–bone interfaces and substantial bone absorption. A HIFU system can safely treat tumors as small as 3 cm in diameter at depths up to 15 cm, while a planar multisource ultrasound applicator can treat larger tumors [32]. Several studies have explored the combination of HIFU-induced hyperthermia with heat-triggered SDDS. A recent Phase I clinical trial in liver cancer patients (NCT02181075) reported promising initial results [33]. HIFU’s precise control, combined with MRI-guided thermometry, enables accurate temperature regulation of tumors. Additionally, two animal studies showed complete tumor regression using this method [34,35].

Medical imaging systems (such as ultrasound, computed tomography, or magnetic resonance imaging) are used to apply internal hyperthermia to tumors, ensuring localized heat within the tumor [36]. Over the past few years, several internal hyperthermia devices have been tested, particularly ultrasound and microwave-based devices [37]. Typically, internal ultrasound devices use multiple transducers to generate localized heating by positioning probes or catheters near the targeted tissue region. These devices have been clinically investigated for the treatment of prostate cancer and benign prostatic hyperplasia through hyperthermia [38,39]. In recent years, microwave-based devices have been developed that offer precise heating and integrate MR thermometry to provide real-time temperature monitoring, which is critical to heat-triggered drug delivery systems (SDDSs) [40].

### 2.2. Hyperthermia-Induced Drug Release

Temperature is an effective trigger for targeted drug delivery, and nanocarriers sensitive to temperature and hyperthermia are used as additional stimuli. Through smart drug delivery systems, nanoparticles can release their cargo in response to environmental temperature change at the diseased site when combined with mild hyperthermia [23]. Figure 2 illustrates how mild hyperthermia enhances nanoparticle delivery to tumors for non-temperature-sensitive (a) and temperature-sensitive (b) nanoparticles. Mild hyperthermia induces greater extravasation and a more uniform distribution of nanoparticles throughout the tumor, due to increased regional blood flow, vasodilation, and enhanced vascular permeability. Additionally, the combination of temperature-sensitive nanoparticles results in an increased cellular release of free drugs after the nanoparticles are heated. This enables nanoparticles to accumulate within tumor tissue, thereby improving drug absorption [14].

Hyperthermia-triggered drug release offers key advantages over pH-, enzyme-, or antigen-based systems because it allows external, adjustable control tailored to individual patient needs [41]. Unlike light-based methods, hyperthermia can be noninvasively applied to deep tissues, and recent advancements enable precise image-guided targeting and real-time temperature monitoring.

Ablative hyperthermia, which heats tumors to the point of tissue destruction, is most effective for well-localized tumors away from critical organs. This intense heating reduces blood flow, intensifies thermal damage, and enhances tumor ablation [42]. When combined with temperature-sensitive drug delivery, it enhances tumor eradication, particularly in regions where heat alone is insufficient for treatment. In such areas, the drug is still released effectively from thermosensitive nanoparticles, supporting tumor control beyond the ablated zone (as shown in clinical trial NCT00617981) [43,44]. In contrast, when tumors are irregularly shaped or near vital tissues, mild hyperthermia (39–42 °C) is safer. Although these lower temperatures do not directly damage cells, they are sufficient to trigger the release of drugs from thermosensitive nanoparticles. This method concentrates the drug locally at the tumor site, reducing systemic side effects and enhancing drug penetration in the heated area [45].

## 3. Thermosensitive Nanoparticles

In recent years, pharmaceutical nanocarriers, including micelles, hydrogels, dendrimers, and liposomes, have garnered significant attention due to their potential to enhance drug efficacy in vivo. A nanocarrier possesses intrinsic properties, including solubility, in vivo stability, biodistribution, and prolonged circulation in the bloodstream. It can also deliver drugs to specific sites and respond to changes in the local environment. Temperature-responsive nanoparticles (TNPs) with prolonged circulation times offer several advantages, including the ability to accumulate at precise tumor locations either passively or actively, which enables controlled drug distribution and minimizes side effects. They can also release drugs gradually in response to environmental temperature changes to improve therapeutic efficacy [46]. Figure 3 illustrates commonly used nanocarriers that are sensitive to changes in temperature.

To produce TNPs, monomers, polymeric solutions, or macrogels may be used. Hoare et al. [47] considered three approaches to synthesis: homogenous nucleation, emulsification, and complexation. With homogeneous nucleation, a uniform polymeric solution is required that contains a monomer and a crosslinker, allowing for straightforward functionalization while introducing a greater variety of monomers. As a result of polymerization, the sodium dodecyl sulphate (SDS) concentration affects the thermoresponsive properties of the resulting particles. During emulsification, also known as mini-emulsion polymerization or inverse emulsion polymerization, hydrophilic monomers are dispersed in an aqueous phase before being polymerized in a continuous non-aqueous medium, producing thermosensitive particles. TNPs, approximately 100 to 200 nanometers (nm) in size, can penetrate biological barriers, making it easier to deliver substances and medications to target areas. The small size of these materials allows them to respond rapidly to external changes, particularly for practical applications at critical solution temperatures (CSTs). In addition, TNPs are characterized by a large specific surface area relative to their size, leading to an increase in active binding sites, which enhances uptake and biological permeability. Furthermore, homogeneous populations demonstrate superior reaction kinetics to CST alterations due to their uniform size distributions [19].

The organic TNPs are categorized into two main parts: polymer and lipid-based nanocarriers. Polymeric micelles, hydrogels, dendrimers, and liposomes are the most common nanocarriers used as thermoresponsive DDSs.

### 3.1. Polymer-Based Thermosensitive Nanocarriers

These nanocarriers respond actively to internal and external thermal changes by incorporating thermosensitive polymers into nanocarrier systems. This affects their conformation, solubility, and hydrophilic–hydrophobic balance [46]. A TNP comprises special polymers that protect the drug during systemic circulation and enable the drug to reach the locally heated tumor. When heated, the TNPs contain both hydrophilic and hydrophobic subunits that undergo reversible phase transitions. Hydrophilic subunits form hydrogen bonds with water molecules, thereby maintaining the polymer’s hydration and maintaining its coil-shaped structure. When you dissolve this thermosensitive polymer in water, it forms a single or homogeneous phase. The temperature change will induce this thermosensitive polymer to change its structure from a coil to a globule. As a result of globule conformation, hydrophilic subunits are separated from water molecules, resulting in dehydration of the polymer. This is where the biphasic or heterogeneous phase will arise (Figure 4) [48].

The polymer is first dissolved in a solvent and heated to form a single-phase solution. Polymer crystallization occurs during slow cooling, causing the system to become heterogeneous and produce a cloud-like condition known as cloud points. A phase diagram is formed by plotting the cloud points on a Cartesian graph. This illustrates the boundary between monophasic and biphasic states, typically represented by a parabolic curve. The highest cloud point, also known as the Upper Critical Solution Temperature (UCST), or the lowest cloud point, also known as the Lower Critical Solution Temperature (LCST), is an important feature of a phase diagram, depending on whether it represents the maximum or minimum temperature, respectively [48].

When thermosensitive polymers are dissolved in aqueous solutions, the LCST phenomenon occurs, resulting in a phase separation caused by the collapse of the polymer chain and the expulsion of water. LCST and UCST divide thermoresponsive polymers into groups based on hydrophilic/hydrophobic monomer ratios. They are then classified based on their thermosensitive mechanisms or chemical compositions [46]. According to the CST, thermoresponsive polymers can be divided into two categories: those with an LCST that become water-soluble and form homogeneous systems below a certain temperature, undergoing a coil-to-globule transition upon heating. Meanwhile, those with a higher UCST undergo a coil-to-globule transition when cooled below a certain temperature [48].

Poly(N-isopropyl acrylamide) (PNIPAAm) is a significant LCST polymer. At temperatures below 32 °C, it forms soluble chains due to hydrogen bonding with water molecules, but at temperatures above 32 °C, it contracts due to water loss. By copolymerizing PNIPAAm with different monomers, positive, negative, or neutral monomers can create diverse structures that modulate the temperature response. Another thermosensitive particle, Pluronic F127, comprises amphiphilic ABA-type triblock copolymers. Due to thermally induced collapse and hydrophobic interactions, this type of polymer offers reversible gelation, biocompatibility, and prolonged drug residence times [19]. Polyethylene glycol (PEG) is a polymer that dissolves in water and has low toxicity. It is useful in enhancing the solubility and thermal stability of hydrophobic drugs. Polysaccharides are natural polymers that can be utilized in various biomedical fields, including drug delivery and tissue engineering. Poly(N-(2-hydroxypropyl) methacrylamide) (pHPMA) offers multifunctionality for the formation of polymeric micelles. These micelles have hydrophilic shells and hydrophobic cores. Poly(amino acids) and their derivatives are biodegradable and biocompatible, making them useful in forming polymeric micelles. In contrast to traditional surfactants, polyethers allow the distribution of active ingredients with minimal toxicity. Poly(L, D-lactide, PLA) is a polyester with customizable mechanical properties and biodegradability, making it useful for various applications [49]. Table 1 summarizes the most commonly used temperature-responsive polymers, along with their key properties and the polymerization methods employed in various applications.

#### 3.1.1. Thermosensitive Micelles

Molecules are a type of nanocarrier that can self-assemble into specific systems, increasing stability and unique properties. The self-assembly of surfactants and polymers results in micelles and vesicles. At the critical micelle concentration (CMC), polymers form micelles with hydrophobic tails and hydrophilic heads [54]. Micelles are small, colloidal particles ranging from 5 to 100 nm formed in water by amphiphilic agents [55]. The size of micelles significantly contributes to EPR within tissues, which causes them to pass easily through leaky vasculature [56]. Depending on the polymer’s composition and solvent conditions, micelles can assume various shapes, including spheres, tubules, or inverse micelles. Dilution or solvent evaporation may be used to prepare micelles. Micelles enclose hydrophobic drugs in their cores while interacting with water molecules through their hydrophilic shells. When the concentration of amphiphilic polymers exceeds the CMC, they form spherical micelles. As shown in Figure 5a, the hydrophobic tails are gathered in the core, while the hydrophilic heads form a shell around the core [54].

An inverse micelle can be formed in non-aqueous solutions with hydrophilic heads pointing inward and hydrophobic tails pointing outward (Figure 5b). It is possible to make mixed micelles from combinations of surfactants or polymers, which are useful for various applications, including drug delivery [54].

Using polymer micelles (PMs) for drug delivery offers many benefits. Their advantages include reduced side effects, targeted delivery, gradual drug release, escape from body defenses, intracellular effects, and stability [57]. The stability of micelles in physiological environments relies on their CMC, hydrophobicity, and intra-micellar interactions. Several methods are being used to improve PMs’ drug-loading efficiency and physical stability, such as adding fatty acids to the core, altering the core structure, and covalent crosslinking [49]. PMs have a nanoscale size and a core–shell structure that protects against oxidation and can encapsulate a wide range of drugs. The discovery of PMs opens up new opportunities for pharmaceutical delivery, which may pave the way for a brighter future [58,59].

PMs have a hydrophobic inner core of polypropylene oxide (PPO) and a hydrophilic outer layer of polyethylene oxide (PEO) that protects the micelles from unwanted interactions [57]. The PMs can also be modified with ligands to target specific tissues, cell-penetrating moieties to accumulate intracellularly, and stimuli-sensitive groups to release drugs in response to stimuli. Hydrophilic heads and hydrophobic tails align at low concentrations at interfaces such as liquid/air or liquid/oil, reducing surface tension. At high concentrations, micelle-like structures form on the surface, covering the entire surface [60].

Two stimuli-sensitive polymer micelles (SSPMs) exist: intrinsic/endogenous and extrinsic/exogenous. Intrinsically thermosensitive PMs are fully soluble and extended below the LCST. Above the LCST, they become water-insoluble. Generally, PMs are circulated through tumor tissues heated above their LCST. As a result, the outer shells of micelles become hydrophobic, enabling them to be absorbed into cells through hydrophobic interactions. By accumulating in cancerous tissues, anticancer drug-loaded micelles kill cancerous cells. PNIPAAm is the most commonly used thermoresponsive polymer with an LCST of 32 °C. When PNIPAAm is dissolved in water, it undergoes a reversible phase shift. Extrinsic thermosensitive PMs allow sustained drug release in targeted tissue areas. External stimuli trigger the release of drug carriers by causing a high temperature. In response to an external stimulus, thermosensitive polymer swelling (hydrophobic-to-hydrophilic conversion) releases heat, simultaneously releasing therapeutic effects associated with chemotherapy [60].

Han et al. [61] studied thermosensitive micelles, which undergo hydration and elongation when hydrated below the LCST but collapse when dehydrated above the LCST. It is known that copolymers like N-acryloyl-Ala-methylester (NAAMe) and N-acryloyl-βAla-methylester (NAβAMe) form micelles whose LCST increases with m value (from 27 °C to 40 °C). To achieve precise thermosensitivity and reversibility, the rigidity of the micelle core ensures stable encapsulation and release of the drug. In this study, deoxycholic acid is conjugated to monomethoxy poly(ethylene glycol) (mPEG) at a temperature above its LCST to reduce the initial burst release of the drug. The micelles prepared with deoxycholic acid as the rigid core and mPEG as the shell demonstrated consistent LCSTs and controllable transition temperatures. It was observed that the micelles exhibited a high degree of reproducibility and reversibility during their thermal transitions. This study demonstrated the preparation of thermosensitive micelles with rigid cores using small-molecular-weight PEG with an LCST of 30−35 °C.

De Moraes et al. [62] synthesized linear AB-type amphiphilic diblock copolymers, Poly(ε-caprolactone)-b-poly(N-isopropyl acrylamide) (PCL-b-PNIPAAm), using controlled ROP and RAFT. These copolymers were then used to create self-assembling micelles that can be utilized for drug delivery in aqueous solutions. Pyrene fluorescence was used to determine the CMC. The results showed that lower CMC values were observed with increasing hydrophobic polycaprolactone (PCL) segment length, while higher CMC values were observed with increasing hydrophilic PNIPAAm segment length. Successful synthesis was confirmed, and the copolymers showed reduced crystallinity compared to PCL-based macrochain transfer agents. As PCL and/or PNIPAAm segments become longer, the size of the micelle increases, while the LCST decreases as PCL segments become longer and increases as PNIPAAm segments become longer. The use of copolymers to control micelle properties could prove to be useful for delivering controlled drugs.

Wang et al. [63] synthesized thermoresponsive amphiphilic block copolymers, Poly(vinyl stearate)-block-poly(N-vinylcaprolactam) (PVS-b-PNVCL) and Poly(vinyl laurate)-block-poly(N-vinylcaprolactam) (PVL-b-PNVCL), using RAFT polymerization under microwave irradiation. The goal was to explore the impact of fatty acid type and chain length on micellization. The block copolymers exhibited lower CMC values compared to PEG–lipid conjugates, indicating enhanced stability in the bloodstream. The study also investigated the temperature-triggered release of doxorubicin (DOX) from PVS-b-PNVCL micelles, revealing a two-phase release pattern with sustained slow release up to 72 h. The biocompatibility and potential for biomedical applications of the micelles were demonstrated in cancerous and normal cell lines. Additionally, the study examined the cytotoxicity and cellular uptake of DOX-loaded micelles, showing promising outcomes for drug delivery.

Manna et al. [64] synthesized a copolymer, Alg-g-PNIPAAm, using the RAFT technique. This copolymer can form drug-carrying micelles that respond to changes in pH and temperature. These micelles have been stable over time and have shown promise as a drug delivery system. Furthermore, the copolymer is non-toxic to certain cell lines and has demonstrated sustained drug release behavior.

Du et al. [65] developed SP94-modified UCST polymeric micelles that co-encapsulate the small-molecule targeted drug Lenvatinib (LEN) and the near-infrared second-region (NIR-II) fluorescence probe IR-1061-AcD (SPLI). The polymer could self-assemble into micelles in an aqueous solution and displayed a temperature-sensitive property with a UCST of 43 °C. The micelles depolymerized above 43 °C, and their disruption led to the aggregation of LEN and IR-1061-AcD. SPLI showed NIR-controlled drug release behavior and promoted temperature increases under NIR laser irradiation. Furthermore, SPLI demonstrated strong signals in the tumor region and high accumulation at the tumor site in mice. In vivo testing demonstrated that SPLI rapidly increased the tumor site temperature to approximately 43 °C, facilitating rapid drug release and photothermal therapy. Additionally, SPLI plus laser treatment enhanced the antitumor immune effects and inhibited the growth of metastatic tumors.

Wang and colleagues [66] investigated the influence of different aromatic substituents on the self-assembly and drug delivery properties of thermoresponsive PCL diblock copolymers. Their findings revealed that polymers with less flexible aromatic groups displayed sharper phase transitions and LCST. These copolymers formed stable micelles and exhibited a faster drug release rate. Additionally, the study confirmed the biocompatibility of the polymers and emphasized the impact of LCST values on drug release and cellular uptake.

Zlotnikov et al. [67] conducted a study to improve cancer treatment using smart micellar systems and adjuvants to overcome multidrug resistance in tumor cells. They developed chitosan-based polymer micelles with fatty acids and eugenol, showing promise for targeted drug release in tumors. The micelles are pH- and temperature-sensitive, releasing the drug in an accelerated and controlled manner at pH 5.5 and 37–42 °C. Micelles containing eugenol increased doxorubicin (DOX) accumulation in cancer cells by over three times, offering promising prospects for effective cancer treatments.

Bagheri-Meyabad and colleagues [68] synthesized a copolymer using the RAFT polymerization method. This copolymer has a very low CMC, indicating its likelihood to form micelles in contact with water above a specific temperature known as the LCST. The presence of PEG in the copolymer raised the LCST, making it adjustable to the body’s temperature. The drug loading and entrapment efficiency in the prepared micelles were found to be higher compared to similar nanomicelles containing at least one NIPAM component. The in vitro drug release performance of PMs/ZnPP was evaluated at various temperatures in a physiological environment. Below the LCST, a higher percentage of drug release occurred over a short period, while above the LCST, the drug release was slower and sustained. Notably, temperature had a significant influence on the drug release from the prepared micelles. The cytotoxicity of the prepared micelles and the drug-loaded micelles on prostate cancer cells was also evaluated, revealing that drug loading increased cytotoxicity and induced apoptosis in the cancer cells. The study concluded that temperature-responsive polymeric micelles could effectively control drug release, potentially enhancing therapeutic effects and minimizing side effects on healthy tissues.

Qiu et al. [69] have successfully developed a thermal-responsive triblock polypeptoid that demonstrates self-assembly into micelles when dissolved in water. The research investigates the polymerization degree, the impact of concentration on assembly, and the encapsulation and staged release of ciprofloxacin drugs in micelles through hydrophobic interactions. Additionally, the study examines the impact of higher temperatures and concentrations on micelle stability and the effectiveness of drug encapsulation. The study also examines the influence of temperature on polypeptoid micelles, revealing significant effects of different block lengths and proportions on particle size and temperature sensitivity. Moreover, the research establishes the good biocompatibility and low cytotoxicity of the polypeptoids. Lastly, the cluster peptide block copolymers demonstrate effective drug encapsulation for sustained release, resulting in a slower drug release rate compared to unwrapped drugs. Table 2 summarizes the most recent in vitro studies that used thermosensitive micelles for cancer treatment. The components, payload, and cancer cell line used in these studies with temperature-triggered release are listed below, along with the main findings.

#### 3.1.2. Thermosensitive Hydrogel

Hydrogels are polymeric networks that retain water while remaining insoluble in nature. They are classified based on bonding nature, physical appearance, mechanical behavior, and swelling ability. Some hydrogels function as smart devices, responding to various stimuli, including temperature, pressure, pH, and enzymes. One of the most prominent characteristics of smart hydrogels is their ability to respond to temperature variations. They must be within a specific temperature range to change from liquid to gel. Drug delivery applications benefit from the mechanical stability, thermo-transition behavior, biocompatibility, and ease of formulation of these materials. There are different thermosensitive groups within these hydrogels, some responding positively to temperature changes (i.e., UCST) and others negatively (i.e., LCST). It is important to note that LCST polymer-based hydrogels are hydrophobic, while UCST polymer-based hydrogels are hydrophilic. The solubility of these eco-friendly hydrogels can be adjusted by varying the amounts of LCST and UCST polymers. These hydrogels provide prolonged drug release at physiological temperatures, making them ideal for drug delivery applications. A thermosensitive hydrogel is defined primarily as a material composed of natural or derived thermosensitive polymers, synthesized through traditional techniques such as crosslinking and polymerization. Unlike natural hydrogels with peptide and glucose ring structures, derived hydrogels have chemical modifications. Smart hydrogels differ from traditional ones due to their phase transition from sol to gel, which is determined by the copolymer’s functional groups and the system’s energy. The mechanical properties of hydrogels, including their viscoelastic and rubber-like nature, are well-known. However, what sets thermosensitive hydrogels apart is their superior adhesion, which is attributed to the polymers used, making them inherently sticky. This unique property has practical implications, allowing for better drug delivery. The phase transition from sol to gel and vice versa, characterized by defining properties such as the UCST or LCST, results from the dissociation of weak bonds. There are three mechanisms by which these thermosensitive hydrogels release drugs: diffusion, swelling, and erosion, with diffusion (the movement of particles from a high to a low concentration) being the primary method. The size of the drug molecule and the temperature are factors that influence drug release kinetics. A swelling or erosion mechanism also involves macromolecule chains dissociating and reversible or irreversible interactions between polymer ions in response to stimuli [70].

Various natural and synthetic materials have been explored to develop thermosensitive hydrogels. Synthetic amphiphilic block copolymers, such as diblock and triblock copolymers, can self-assemble into micellar aggregates at a critical aggregation concentration (CAC). The most commonly used triblock copolymers contain hydrophilic (A-blocks) and hydrophobic (B-blocks) groups arranged in an ABA or BAB pattern. A-blocks provide biocompatibility and water solubility, whereas B-blocks provide hydrophobic drug loading. A unique property of these copolymers is their ability to transition from free-flowing solutions at room temperature to gels at body temperature. An injectable thermosensitive hydrogel offers a promising biomaterial with an LCST. At physiological temperatures, it forms a hydrogel that allows easy encapsulation and injection of therapeutic agents in solution. Gel formation in situ may enhance drug retention at injection sites, act as drug depots, and reduce systemic toxicity. Combining thermosensitive hydrogels with nanoparticle-based formulations can increase drug loading capacity and slow drug release rate. An example of a BAB triblock pattern is the PLGA-b-PEG-b-PLGA polymer, where PLGA represents poly(lactic-co-glycolic acid) and PEG stands for poly(ethylene glycol). These polymers form a flower-like micelle structure in water. In Figure 6, gelation occurs when the critical gelation concentration (CGC) and critical gelation temperature (CGT) are reached, and it is caused by hydrophilic bridges formed by unfolded polymer chains [71].

Patel et al. [72] studied the feasibility of creating a thermosensitive hydrogel for drug delivery. They used poloxamer 407, a well-known triblock copolymer, to design the hydrogel system. They synthesized various triblock copolymers based on the ABA architecture, using PEG and PCL with different molecular weights. They developed a triblock copolymer based on PEG-PCL-PEG and modified PCL with hydrophilic PEG. This modification led to the formation of a stable micellar structure or hydrogel network, which facilitated drug delivery. The researchers in this study created a collection of triblock copolymers and analyzed their ability to withstand heat and be injected through a syringe. The PEG-PCL-PEG copolymers showed a decrease in viscosity as the temperature increased. This implies that they may not form a gel or undergo a sol–gel transformation at body temperature (37 °C). On the other hand, the PCL-PEG-PCL copolymers formed gels when exposed to room or body temperatures. There was an emphasis on triblock polymers that exhibit a sol–gel transition at body temperature. The PEG-PCL-PEG copolymers remained soluble in water and showed lower viscosity at physiological temperatures, indicating an unlikely sol–gel transformation. The release studies were conducted using diclofenac sodium as a model drug. In the absence of the hydrogel, about 95% of the drug would have been released within two hours. However, when incorporated into a thermosensitive hydrogel, only about 90% of the drug was released over a 14-day period, with a slower release observed after 10 days. This study demonstrated the potential of the hydrogel system to sustain drug release, thereby reducing the need for frequent doses. Additionally, this hydrogel system has unique crosslinking properties that can increase drug loading, entrapment, and targetability. This can overcome challenges associated with therapeutic agents, such as short half-lives, high molecular weights, and high hydrophilicity.

Pluronics (Poloxamers) are a type of copolymer comprised of polyethylene oxide–polypropylene oxide–polyethylene oxide (PEO-PPO-PEO) triblock. These copolymers have several advantages: they are safe, adhere well to biological surfaces, are stable, and form gels at low concentrations when exposed to body heat. The gelation properties of Pluronics can be adjusted by blending them with other components or adding them to other components. Bana Shriky and their team conducted innovative experiments that combined rheological analysis with small-angle X-ray and neutron scattering to investigate the effects of different concentrations of Pluronics (ranging from 1% to 35%). They found that concentrations above 15% underwent a reversible thermal transition from micellar liquids to shear-thinning physical gels. This discovery offers potential applications for injectable drug delivery [73].

PLGA-based thermosensitive hydrogels offer precise spatial and temporal control over delivery, making them ideal for various drug delivery applications. PLGA, a biodegradable, adsorbent, and biocompatible copolymer composed of lactic acid and glycolic acid, is an FDA-approved material suitable for drug delivery. Due to their degradability, drugs can be transported in these carriers, which allows the controlled release of drugs at optimal doses. PLGA-based hydrogels can deliver hydrophobic and hydrophilic molecules to tissues [74].

Hydrogels such as the thermoresponsive PLGA-PEG-PLGA encapsulating DOX undergo a sol–gel transition at body temperature. In contrast to free DOX, this formulation forms a gel that persists for 20 days after injection into rats, releasing DOX at sustained levels with minimal toxicity. Nevertheless, DOX loading into injectable hydrogels should be adjusted for potential viscosity changes. Although PLGA-based hydrogels have many advantages, they also have limitations. PLGA nanoparticles, for instance, do not have specific interactions with cells or proteins, which prevents targeted drug accumulation in tissues, and their production costs can be high. Furthermore, drug–polymer interactions influence sustained drug release from PLGA-based carriers. Various PLGA-based thermoresponsive hydrogel formulations have been developed for various biomedical applications, including cancer therapy, with consideration given to mechanical properties, pore size, and water absorption factors. Hydrophilic PEG and hydrophobic PLGA can be combined in diblock copolymers to regulate hydrogel properties in vitro and in vivo. In drug delivery, thermosensitive hydrogels made from triblock copolymers are preferred because they can form gels for various tissues without being as water-soluble as diblock copolymers. PLGA-PEG-PLGA, for instance, is commonly studied for its biocompatibility, biodegradability, and non-toxicity. Solving challenges such as rapid drug release from hydrogels using triblock copolymers, which facilitate rapid gel formation with minimal burst release, is possible. It is crucial to adjust the phase transition temperature of thermosensitive hydrogels, as reversible and irreversible transitions offer different advantages. A photocrosslinkable PEG-PLGA-PEG study with Irgacure 2959 demonstrated that irreversible chemical crosslinked hydrogels have superior mechanical properties. This allowed for rapid conversion to a gel and caused the formation of irreversible hydrogels [74].

Kim J et al. [75] have developed a thermosensitive hydrogel system composed of gelatin and Pluronic F127. This system releases a protein called cytotoxic T-lymphocyte-associated (CTLA-4), known to boost the immune system, and a nitric oxide donor that can enhance the immune system’s ability to fight cancer. The hydrogel system can effectively retain therapeutic agents at the tumor site and release them when triggered by the tumor microenvironment (TME). This creates in situ micelles that the lymphatic system can use, making it a promising platform for clinical use. By enhancing nitric oxide’s immune-modulating properties, this innovative platform holds promise for clinical use, potentially enhancing immune checkpoint blockade therapy.

Laurano et al. [76] present a promising approach for efficient drug delivery based on hydrogels composed of micellar structures. To address challenges such as burst drug release and difficulty loading hydrophobic drugs into hydrophilic networks, the researchers synthesized thermosensitive micellar hydrogels from an amphiphilic poly(ether urethane) (PEU) made from Poloxamer R 407. Drug molecules are bound to polymeric chains to mitigate burst release, and the degree of crosslinking is controlled in polymer networks. Polymer chains are amphiphilic, allowing them to interact with hydrophilic and hydrophobic drugs to form micellar hydrogels. Furthermore, Poloxamer 407-based PEU gels offer a higher degree of gelation, improved mechanical strength, and enhanced aqueous stability compared to native Poloxamer R 407 hydrogels. Hydrogels based on poly(ether urethane) can simultaneously encapsulate and release multiple therapeutic agents, enhancing therapeutic efficacy by targeting co-delivery into specific tissues and organs. Table 3 summarizes the most recent in vitro and in vivo studies that have utilized thermosensitive hydrogels for cancer treatment. The table includes the components, payloads, and cancer cell lines used in these studies, as well as the main findings related to drug release.

#### 3.1.3. Thermosensitive Dendrimers

In the 1950s, scientist P. Flory introduced the concept of hyperbranched three-dimensional dendritic polymers. The first synthesis of dendrimer structures occurred between 1980 and 1990; a period during which researchers like Fritz Vogtle and Tomalia started exploring dendrimers, which led to the production of low-molecular-weight hyperbranched polymers [86]. Dendrimers are nano-sized, symmetric molecules that have a well-defined tree-like structure made of branches and arms. They consist of symmetric units that branch from a central core, which can be a small molecule or a linear polymer. Figure 7 illustrates the molecular structure of dendrimers, highlighting their highly branched, tree-like architecture. Dendrimers are recognized by their ability to function at their end groups, which allows for the tailoring of chemical and biological properties. As the dendrimers increase in generation, they become larger and more globular in shape, which enhances their application in drug delivery and diagnostic imaging [87]. For example, a study highlighted the unique structure of the 4th generation poly(amidoamine) (G4.0 PAMAM), a higher-generation dendrimer that encloses drugs in internal spaces with considerable stability. The research discussed preparing an active G4.0 PAMAM complex with doxorubicin hydrochloride and monitoring its properties. The study concluded that because the dendrimer was of a higher generation, it provided stability and delivered Doxorubicin, enhancing its therapeutic effect [88]. In contrast, lower-generation dendrimers have a limited capacity to hold medications, as demonstrated in a study examining the 2nd-generation poly(amidoamine) (G2.0 PAMAM) as a carrier for drug delivery. G2.0 dendrimers were able to deliver Doxorubicin, but their small size and limited functional groups decreased targeting properties and loading capacity compared to higher generations [89].

The synthesis of dendrimers occurs through two major approaches: divergent (Figure 8a) and convergent (Figure 8b). In the divergent approach, synthesis begins with a central core to which monomers with reactive groups are subsequently added. This is accomplished in a stepwise manner, resulting in progressively generated dendrimers. This method is effective for producing high yields of dendrimers, but is often accompanied by challenges such as side reactions, structural defects, and the requirement of large amounts of reagents, which can complicate the purification process. On the other hand, the convergent approach constructs dendrimers by first starting from the peripheral branches, or dendrons, before attaching them to a central core. This method can decrease the incidence of defects and facilitate purification because it avoids the need for excess reagent. However, the hindrance arising from the pact branches can compromise the production of higher-generation dendrimers.

The two main dendrimers that are commercially available are Poly (Propylene Imine) PPI, synthesized by the convergent method, and Poly (Amido Amine) PAMAM, synthesized by the divergent method. Recent innovations in synthesis have led to the development of new families of dendrimers, such as Tectodendrimers and peptide dendrimers. Tectodendrimers are composed of high-generation cords and low-generation shells, which exhibit enhanced properties such as solubility and biocompatibility. Peptide dendrimers can be engineered to display ligands, enhancing their utility for drug delivery [90].

Temperature-sensitive dendrimers are a type of dendrimer with temperature-adaptive behavior. Changing their solubility or structure in response to temperature changes [91]. They are characterized by LCST and a UCST phenomenon, where the dendrimers switch between soluble and insoluble states. This makes them valuable in drug delivery and release to target sites [92,93]. Type LCST thermosensitive PAMAM dendrimers and dendritic polymers, which are helpful for drug delivery and substance separation, have been created by modifying them with different substances, oligo (ethylene glycol), N-isopropyl groups, and elastin-like peptides [94,95]. Phenylalanine (Phe)-modified PAMAM dendrimers are also thermosensitive; however, their terminal groups determine the degree of sensitivity. Above pH 6, the PAMAM-Phe amino-terminal Phe-modified PAMAM dendrimer demonstrated LCST-type thermosensitivity [96]. On the other hand, at acidic pH, the carboxy-terminal Phe-modified dendrimers displayed UCST-type thermosensitivity [97]. A study by Wu et al. [98] examined temperature-sensitive dendrimers based on oligo (ethylene glycol) (OEG), which are produced up to the fourth generation through amidation and alkyne–azide cycloaddition processes catalyzed by Cu(I). Because of their increased hydrophobicity, these dendrimers exhibit thermosensitivity, with a decreasing LCST as production increases. For instance, at 0.25 mg/mL, the LCSTs for G2, G3, and G4 in deionized water were 31 °C, 24 °C, and 17 °C. PEG chain modifications (G4′ and G4″) improved solubility and modified LCSTs; because of its strong hydrophilicity, G4″ shows no LCST below 100 °C. The peripheral modifications of G4 are different in the two versions: G4′ has unmodified end groups, which makes it less soluble and smaller (~4.6 nm), whereas G4″ has longer PEG chains, which make it larger (~5.8 nm), more hydrophilic, and able to circulate for longer periods of time and accumulate more tumors. In terms of tumor targeting and penetration, G4″−GEM performed better than G4′−GEM and PAMAM dendrimers when coupled with gemcitabine (GEM). Micro-PET and MSOT imaging revealed that G4″−GEM had the highest tumor uptake (8.7% ID/g) and superior penetration into poorly vascularized regions. Due to these characteristics, G4″−GEM significantly inhibited tumor development (67.7%), while PAMAM−GEM and free GEM did so at 47.2% and 39.8%, respectively. The promise of OEG-based dendrimers, especially G4″, as thermosensitive drug delivery vehicles for improved cancer treatment is demonstrated by these findings [98].

By altering the terminal groups of dendrimers with isobutyramide (IBAM), a group that is recognized for its thermosensitivity, Haba et al. [99] investigated making dendrimers temperature-sensitive. LCST behavior was demonstrated by IBAM-functionalized PAMAM and PPI dendrimers, whose solubility varied with temperature. Because of the higher peripheral IBAM group density, the LCST for IBAM-PAMAM dendrimers declined dramatically as dendrimer generation increased. For instance, IBAM-G5 had an LCST of 22 °C, but IBAM-G4 and IBAM-G3 had LCSTs of 29 °C and 36 °C, respectively. Solution parameters like pH and concentration also had an impact on the LCST. Because of their smaller size and higher IBAM density, PPI dendrimers functionalized with IBAM, including DAB-Am-64 and DAB-Am-32, showed lower LCSTs than their PAMAM counterparts, with LCSTs of 27 °C and 31 °C, respectively. These results demonstrate how dendrimers can be directly made thermosensitive, which increases their adaptability for stimuli-responsive applications. Another study, conducted by Zheng et al. [100], utilized amphiphilic dendrimer-like copolymers with a polystyrene core and a poly(ethylene oxide) (PEO) shell, which function as thermoresponsive unimolecular nanoreactors. The LCST of the PEO segments, which is 65 °C, causes the polymers produced by olefin metathesis and anionic polymerization to exhibit extreme temperature sensitivity. As unimolecular micelles, the dendrimers encapsulate hydrophobic molecules and promote organic reactions below LCST. For example, they significantly boost the reaction rates of benzyl halide reactions in aqueous media when compared to traditional systems. The dendrimers agglomerate and lose their catalytic activity over LCST. The alternating activation and deactivation in the hydrolysis of benzyl chloride demonstrates how this heat responsiveness allows for precise control over processes. The potential of dendrimers in environmentally friendly chemical processes is highlighted by their reusability and ability to maintain catalytic performance across multiple cycles.

Tamaki et al. [91] examined the synthesis as well as the pH- and temperature-sensitive behaviors of PAMAM-Suc-Phe and PAMAM-Phe-Suc, which are carboxy-terminal Phe-modified PAMAM dendrimers that were created by changing the order of interaction between succinic anhydride and phenylalanine (Phe). Depending on the solution’s pH, both dendrimers displayed different thermosensitive behaviors. Both dendrimers were transparent at pH 6, but their heat sensitivities varied at lower pH levels. At pH 5, PAMAM-Suc-Phe demonstrated UCST-type thermosensitivity, which became evident when heated, but it did not display LCST-type behavior at any other pH. PAMAM-Phe-Suc, on the other hand, showed both UCST-type thermosensitivity at pH 5.5 and LCST-type thermosensitivity at pH 4, with a pH-dependent transition between behaviors. The two dendrimers showed distinct pH- and temperature-sensitive characteristics despite having similar chemical compositions, emphasizing the influence of structural arrangement on thermosensitivity. According to these results, PAMAM-Phe-Suc can be used to create thermoresponsive materials with adjustable properties that have potential uses in self-assembly, drug transport, and other biological domains.

Using poly(N-isopropylacrylamide) (PNIPAM) and FeRh alloy, Amirov et al. [101] investigated the creation of a thermoresponsive composite for regulated drug release, as demonstrated by the chemotherapeutic agent doxorubicin (DOX). The magnetocaloric effect (MCE) of FeRh alloy was combined with the thermosensitivity of PNIPAM, which is defined by its LCST of about 32 °C. In order to provide precise, non-invasive drug release without the need for conventional heating systems, the composite reacts to external magnetic fields by initiating phase transitions in PNIPAM. In vitro tests showed that the cooling of the composite by the magnetic field activated the LCST, which in turn caused the polymer to change from a dehydrated to a hydrated state, releasing DOX. Spectroscopic investigations verified effective drug release, and mouse embryonic fibroblast biocompatibility experiments revealed satisfactory adhesion and proliferation on the composite surface. The promise of thermoresponsive polymers in targeted cancer therapy is demonstrated by this magnetic field-controlled system. In another study by Sara et al. [102], poly(N-isopropylacrylamide) (PNIPAAM) and poly(amidoamine) (PAMAM) dendrimers were combined to create smart, temperature-sensitive hydrogels. Depending on the temperature, these hydrogels were made to absorb and release medications in a regulated manner. The hydrogels became more homogeneous in structure, more capable of carrying more medicines, and better at retaining water when PAMAM dendrimers were added. The hydrogels swelled and absorbed the medications at low temperatures, but they rapidly shrank and released the pharmaceuticals as the temperature rose above 37 °C, which is nearly body temperature. The unique temperature-sensitive characteristics of PNIPAAM, which alter its structure at about 32 °C, were the cause of this behavior. The researchers used paracetamol as a test medication and discovered that, while the hydrogels absorbed the drug slowly, they released it swiftly at higher temperatures. PAMAM’s addition enhanced medication absorption without appreciably changing the rate of release. Although there is potential for this work in enhanced drug delivery, it was not explicitly evaluated with cancer medications or for cancer therapy. Table 4 summarizes the most recent in vitro and in vivo studies that used thermosensitive dendrimers for cancer treatment. The components, payload, and cancer cell lines used in these studies with temperature-triggered release are listed below, along with the main findings.

### 3.2. Lipid-Based Thermosensitive Nanocarriers

#### 3.2.1. Thermosensitive Liposomes

The first liposomes were discovered in the 1960s by Dr. Alec D. Bangham, a British hematologist at the Babraham Institute, University of Cambridge [106]. Liposomes are small colloidal spherical capsules with multiple lipid bilayers surrounding a water-based, oil, solid, or polymeric core formed by self-assembling amphiphilic lipid molecules, such as phospholipids. The lipid bilayers of liposomes surround an aqueous core with polar head groups oriented towards the inner and outer phases. It is possible to adjust their size and number of layers during manufacturing. Thus, there are three categories of vesicles: small and/or unilamellar (one layer), and multilamellar (multiple layers). Because of this unique structure, liposomes can encapsulate hydrophilic and lipophilic drugs. Large molecules, such as proteins and enzymes, cannot pass through the bilayer (a two-layer structure) due to their low permeability to charged molecules. A protective PEG layer is often added to improve stability in the bloodstream [107]. The structure of liposomes facilitates the efficient loading and delivery of molecules of various solubilities. The aqueous core hosts hydrophilic molecules, the lipid bilayer houses hydrophobic molecules, and the interface between the water and lipid bilayer hosts amphiphilic molecules (Figure 9) [108]. In liposomal formulations containing small-molecule chemotherapy drugs, side effects are reduced, leading to a higher therapeutic effect. Liposomes and other lipid nanoparticles are increasingly used to overcome delivery challenges associated with new pharmaceutical delivery systems [109]. Liposomes are high-performing drug delivery vehicles with flexible structures and numerous advantages, including biocompatibility, safety, and non-immunogenicity. Liposomes can carry large drug molecules and offer a variety of physicochemical properties that can influence their biological behavior. They deliver drugs based on the number of lipid bilayers and rigidity, size, surface charge, lipid organization, and surface modification [108].

Phospholipids’ transition temperature (Tc), which marks the shift from a gel phase to a liquid crystal phase, can affect the fluidity of lipids in a bilayer. Several factors determine the Tc, including the length of fatty acid chains, the saturation level, the ionic strength of the medium, and the nature of the polar head group. Tc is higher in bilayers with longer, saturated chains because of their rigidity and less permeability. Above Tc, the bilayer is in a liquid crystalline phase, with high fluidity but relatively low permeability. Below Tc, the bilayer is in a gel phase with little fluidity and low permeability. When the temperature is around Tc, there are highly permeable interfacial regions between gels and liquid crystals [108]. As shown in Figure 10, the right side represents a disordered, fluid-like bilayer, whereas the left side shows an ordered, solid-like bilayer. The membrane melts as the temperature increases. The two phases coexist around Tc (also known as the melting temperature, Tm, which varies with the length and saturation of the tail group) [110].

A key difference between liposomes and other nanoparticle delivery systems is their ability to adjust their structural and physicochemical properties. This flexibility allows liposomes to be customized within the body and targeted to specific delivery sites. A liposome can be classified according to its composition and functionalization. The design of liposomes has evolved from conventional, stealth, and targeted liposomes to immunoliposomes and stimuli-responsive liposomes [111,112].

Liposomes that react to changes in physiological or biochemical conditions, such as pH, temperature, enzymes, etc., are thought to be stimuli-responsive liposomes. Temperature-sensitive liposomes (TSLs) are probably the most commonly used stimuli-responsive liposomes [108]. Typical TSLs release their payload within 30 min of being triggered at 42–45 °C, while low-temperature sensitive liposomes release the payload within seconds when triggered between 39.5 °C and 41 °C. Figure 11 illustrates the function of thermosensitive liposomes during hyperthermia. The lipid shell is depicted in yellow, while the drug is represented by red triangles. Initially, liposomes undergo EPR effects to cross leaky blood vessels. The large red circle represents the area where hyperthermia is applied. The presence of hyperthermia increases vessel pore size, which facilitates liposomal extravasation. As a result, the drug is likely to enter the tumor tissues and the vasculature, increasing the permeability of the cell membrane. One of the several triggers that induce drug release is mild hyperthermia, which may synergize with high-dose chemotherapy delivery [107].

Several studies have shown that delivering drugs like DOX via liposomes can enhance their therapeutic effectiveness. For instance, the plasma half-life of free DOX (only a few minutes) is significantly longer than that of non-thermosensitive liposomal DOX (10 h) and lysothermosensitive liposomal DOX (~1 h). As a result, liposomal formulations can sustain high blood concentrations, allowing stimuli such as heat to be effective. In addition, thermosensitive liposomes with lysolipids have demonstrated higher drug concentrations in tumors than traditional thermosensitive liposomes [107].

TSLs are primarily administered intravenously and remain in circulation for several hours. Furthermore, TSLs are released in two distinct ways: extravascular-triggered release and intravascular-triggered release. Extravascular triggered release occurs after TSL extravasation, followed by hyperthermia-triggered release. In this way, EPR allows TSLs to accumulate within tumor tissue. Then, the drug is released in the microvasculature through intravascular triggered release as TSLs pass through the heated tumor (where hyperthermia is present), eliminating the EPR effect. As a result, cancer cells absorb the released drug after it extravasates into the tissue. A release of TSL is triggered by hyperthermia after 24 h of accumulation in the tumor, based on the EPR effect. During the intravascular triggered release, applying hyperthermia for 30 min results in drug release, increasing its plasma concentration. This causes the drug to diffuse into the extracellular space and be absorbed by the cells [113].

In a tumor section, blood or plasma containing TSLs enters a supplying artery, crosses tumor capillaries, and exits a draining vein. The length of time that plasma remains within a tumor section is referred to as the tissue transit time (TT). A red blood cell’s movement through capillaries is slower than that of plasma, resulting in a longer stay within a tumor segment. The release of drugs from TSLs and their absorption by tumor tissue can only occur during this period of tissue transit. Upon entering capillaries within a heated tumor region, plasma containing TSLs releases the drug, which is then absorbed by the tumor. Consequently, as plasma moves between the tumor segment’s supplying artery and draining vein, the plasma drug concentration fluctuates along the vasculature [113].

In the early days of thermosensitive liposome formulation, the release kinetics of their payloads were slow, typically requiring minutes to hours for the payload to be released. TSL-encapsulated drugs were often released at temperatures exceeding 42 °C, which had the disadvantage of reducing blood flow and limiting TSL release. Recent studies have shown that fast-release TSL formulations deliver significantly higher levels of drugs than slower-release formulations. The release kinetics of TSL formulations can be compared using a characteristic release time based on linear approximations. Typically, linear approximations can adequately represent the release of TSLs within the initial seconds of their passage through a tumor (transit time) [114].

The plasma stability of a TSL formulation is measured by its initial plasma half-life, which is the time after administration during which the drug remains in systemic circulation. TSL circulating in the blood enters the heated tissue volume, releasing the drug intravascularly during hyperthermia. TSL-encapsulated drug plasma concentrations reflect the amount of intravascular trigger release available. The TSL-encapsulated drug released into the heated tumor directly correlates with the area under the concentration-time curve (AUC) of plasma concentration during hyperthermia. If TSL release kinetics are consistent, a higher AUC results in more drug release. AUC can be optimized by adjusting the timing of hyperthermia, thereby improving drug delivery. TSL plasma stability is challenged by factors such as drug leakage from TSLs at body temperature (37 °C), which conflicts with the rapid release that occurs during hyperthermia. A correlation exists between peak plasma concentrations post-administration and administered dose, typically close to or at the maximum tolerated dose (MTD) for that TSL–drug formulation. Compared to larger animals and humans, rodents with higher MTDs have higher plasma concentrations and tumor drug uptake [115].

To ensure optimal drug release from TSLs, tumor tissue should be uniformly heated to 40 °C and 42 °C. Nevertheless, uniform heating poses a technical challenge, particularly for large tumors. A slight increase in body temperature can trigger premature drug leakage from TSLs, reducing the availability of drugs for release. A longer hyperthermia duration enhanced tumor drug uptake to the point where additional heating provided limited benefit due to lower plasma concentrations of TSL-encapsulated drug. Prolonged hyperthermia increases the uptake of drugs in tumors by increasing the area under the AUC. After a certain point, extending the heating duration provides limited additional benefits. This limitation occurs when the plasma concentration of the TSL-encapsulated drug decreases significantly [116]. Drug delivery is affected by the volume of tissue exposed to hyperthermia, with large volumes depleting the systemic availability of drugs rapidly. In comparison to volumetric methods, such as water baths, focal hyperthermia methods, like infrared lasers, show improved tumor drug uptake. In the event of large tissue volumes being exposed to hyperthermia, such as soft tissue sarcomas or chest wall recurrences after breast cancer, clinical implications may be significant. To ensure effective tumor heating, methods should maximize tumor exposure while minimizing exposure to normal tissue. The use of water bath hyperthermia in rodent studies may reduce delivery effectiveness. To optimize TSL-based therapies, monitoring and controlling heating methods are essential [115]. Furthermore, mild hyperthermia (42 °C for one hour) combined with TSLs increased drug accumulation within tumors and significantly improved treatment effectiveness [117].

Many chemotherapy drugs are available on the market, but some are particularly effective in treating cancer, including Daunorubicin, DOX, Paclitaxel, Gemcitabine, and Cisplatin. DOX is by far the most notable despite its cardiotoxicity, hematological effects, and organ toxicity. A phospholipid-based formulation of DOX (Liposomal-Dox), derived from natural phospholipid-containing vesicles, enhances DOX’s effectiveness while reducing its adverse effects. The drug is available in two forms: pegylated versions (Doxil^®^/Caelyx^®^) and non-pegylated versions (Myocet^®^/Nudoxa^®^/ThermoDox^®^). ThermoDox, a liposomal DOX variant activated by heat, has shown promising results in clinical trials for treating intermediate-risk tumors. Due to its unique structure, DOX is released more efficiently at higher temperatures in targeted areas, leading to reduced toxicity. ThermoDox^®^ contains lipids that enhance membrane permeability and circulation time, including Dipalmitoylphosphatidylcholine (DPPC), Myristoylstearoylphosphatidylcholine (MSPC), and 1,2-distearyl-sn-glycero-3-phosphorthanolamine-N-[amino-(polyethyleneglycol)-2000]. Further enhancement of its efficacy will be achieved by combining thermal therapies such as radiofrequency thermal ablation (RFA) and high-intensity focused ultrasound (HIFU) [118].

As early as two decades ago, Regenold et al. [117] developed ThermoDox^®^, a new formulation that contained lysolipid to facilitate faster and more complete drug release. Based on preclinical studies, ThermoDox^®^ inhibited tumor growth under mild hyperthermia conditions. TSL-mediated drug delivery has long aimed to enhance physiological responses by combining heat-triggered drug release with mild hyperthermia. Preclinical studies have shown that mild hyperthermia increases blood flow and perfusion. Although mild hyperthermia can be effective, factors such as tumor model variability and the applied thermal dose affect its effectiveness, as temperatures above 43 °C can cause vascular contractions. Optimizing thermal dose and treatment timing is crucial to maximize drug delivery via TSLs. Prolonged heating at mild hyperthermia or short heating at higher temperatures can cause vascular blockages that inhibit drug delivery. However, extremely high temperatures (over 50 °C) are effective in directly killing tumor cells, but they can leave behind active cancer cells, resulting in a high recurrence rate. A thermodox product was developed to treat residual cancer cells outside the ablative zone.

Lyso-thermosensitive liposomal doxorubicin, also known as ThermoDox, represents a significant advancement in liposomal drug delivery, particularly in its heat-activated formulation, which has made it a pioneer in human clinical trials. This drug delivery system features several key design elements, including a liposomal carrier with high capacity, a prolonged circulation half-life, and rapid drug release at mild thermal elevations (above 39.5 °C). Synthetic phospholipids such as DPPC, MSPC, and DSPE-mPEG2000 are responsible for these features. PEG prolongs the plasma half-life by stabilizing the molecules, allowing them to accumulate in heated tissues. In the current formulation, over 80% of the liposomal contents are released within 20 s of exposure to the release temperature [119].

Doxorubicin hydrochloride, a potent cytotoxic agent, is enclosed within lyso-thermosensitive liposomes (LTLD). As a result of combined focused ultrasound and LTLD, DOX was significantly more effective at accumulating within tumors than free DOX. This is due to LTLD’s continuous circulation, which allows for prolonged exposure when plasma concentrations are highest. The heating duration must be optimized for maximum payload release at the target site. Despite a leakage rate of 1% per minute in plasma, approximately 50% of LTLD liposomes are still intact after 45 min of targeted hyperthermia. By ensuring sufficient circulation and heating in the target tissue, LTLD has an initial distribution half-life of about one hour [119].

It is not important what type of hyperthermia method is used for LTLD treatment, but rather what temperature and when it should be administered. LTLD has a similar toxicity profile to other anthracyclines, with common non-hematological side effects such as alopecia, nausea, and vomiting. Although cardiotoxicity remains a concern with anthracyclines, LTLD has shown promising safety results when combined with RFA. There are no severe adverse effects, such as congestive heart failure or hand–foot syndrome, associated with LTLD. It can be concluded that LTLD represents a significant advance in targeting drug delivery, especially in conjunction with hyperthermia. It offers better efficacy with more manageable side effects than other treatments [119].

In the study by Aloss et al. [120], DOX encapsulated in LTLD was administered to mice with modulated electro-hyperthermia (mEHT). It was revealed that mEHT + LTLD had the strongest inhibitory effect on tumor growth when compared to mEHT + PLD. This was due to the LTLD’s ability to accelerate the local delivery of DOX. As a result of LTLD, 80% of DOX is released into the bloodstream in a heated tumor, resulting in early accumulation in the tumor. On the other hand, PEGylated liposomal DOX (PLD) delivery relies primarily on the EPR effect, which is a slower process.

Spiers et al. [121] compare intravenous-free doxorubicin delivery to liposomal DOX (ThermoDox^®^) with targeted drug release using extracorporeal focused ultrasound (FUS) to evaluate the effectiveness of enhancing doxorubicin concentration in pancreatic tumors. One arm of the trial will receive doxorubicin alone, while the other will receive ThermoDox^®^ with FUS. As a result of the findings of this early-phase study, ultrasound-mediated hyperthermia could be shown to enhance doxorubicin delivery to pancreatic tumors safely and effectively, opening the door for the combination of multiple anticancer therapies with FUS-induced hyperthermia to treat pancreatic cancer.

J.K. Koehler et al. [122] developed a new type of thermosensitive small multilamellar lipid nanoparticles (tSMLPs) that are highly sensitive to temperature changes. These nanoparticles efficiently release the fluorescent marker calcein in the presence of human serum albumin (HSA). Fabricated through dual centrifugation at high lipid concentrations, tSMLPs are made of dipalmitoyl phosphatidylcholine (DPPC), distearoyl phosphatidylcholine (DSPC), and dipalmitoyl-sn-glycerophosphatidyldiglycerol (DPPG2). They are approximately 175 nm in size with a narrow size distribution (PDI~0.02) and consist of tightly packed lipid layers without an internal aqueous core. These nanoparticles exhibit stability due to hydrogen bonding from DPPG2 and can encapsulate water-soluble drugs with an efficiency of about 25%. Experimental studies using various techniques revealed their unique morphology and temperature-triggered release mechanism. tSMLPs remain stable at body temperature but release their contents rapidly at mild hyperthermia (42 °C), with HSA enhancing this release by stabilizing membrane defects at the lipid phase transition temperature. Cryo-electron microscopy confirmed their distinct architecture, making tSMLPs promising carriers for targeted drug delivery in cancer therapy, especially when used with localized hyperthermia.

F.B. Einolghasi et al. [123] studied the combined effects of hyperthermia and chemotherapy using thermosensitive NPs to enhance drug uptake in cancer cells. They developed a multi-component numerical model to simulate the delivery of focused ultrasound-enhanced doxorubicin with tumor-targeting nanoparticles, enhancing accuracy by incorporating blood coagulation effects and temperature-dependent thermal properties. Their findings showed that employing focused ultrasound-targeted drug delivery systems (FU-TSL DDS) significantly improved drug penetration in tumors, enhancing the effectiveness of chemotherapy while reducing side effects. Increased cellular uptake resulted in a higher proportion of killed cancer cells, thereby reducing the likelihood of tumor recurrence. The study also explored optimal heating conditions for DDS through a parametric analysis of heating power and nanoparticle sizes. It determined effective power levels for achieving desired drug concentrations without overheating. While smaller nanocarriers increased drug concentration, they also incurred higher production costs. Overall, these results emphasize the importance of precise temperature control and heating power to optimize hyperthermia-assisted chemotherapy outcomes.

Table 5 summarizes the most recent in vitro and in vivo studies that have used thermosensitive liposomes for cancer treatment. The table includes the components, payloads, cancer cell lines, and animal models used in these studies, along with the main findings related to drug release.

#### 3.2.2. Thermosensitive Solid Lipid Nanoparticles (SLNs)

Solid lipid nanoparticles are colloidal delivery systems made from solid lipids. SLNs were created in 1991 to enhance storage stability and bioavailability, while protecting integrated medications from degradation. They were designed to address several issues with conventional colloidal carriers, including cytotoxicity, polymer degradation, and the high cost of manufacturing. Compared to traditional carriers, one of the main advantages is a wide surface area and low toxicity, which improves cellular uptake. Additionally, they enhance solubility and bioavailability, enabling sustained drug release and providing a strong foundation for therapeutic applications. The type of lipid matrix utilized and the precise position of the medication inside the formulation are two parameters that affect the drug release profile of SLNs. Because SLNs can enhance the delivery of both lipophilic and hydrophilic bioactive substances, they provide versatile options for achieving targeted and regulated drug delivery [132]. Solid lipid nanoparticles are composed of a crystalline lipid core stabilized by interfacial surfactants, making them promising candidates for drug delivery. The ability to store and release organic molecules is customized by selecting specific lipids and surfactants [133]. Figure 12 illustrates the molecular structure of SLNs.

There are several methods to synthesize SLNs:Solvent-based methods:1.Solvent Emulsification–Evaporation Method: dissolves lipids and drugs in water-immiscible organic solvents, emulsifies in an aqueous faze, and evaporates the solvent, yielding SLNs with nano-sized distribution (100 mL); however, it requires the removal of toxic solvents [134].2.Solvent Emulsification–Diffusion Method: includes saturating water and organic solvents, preparing nanoemulsions, diluting with water, and then disposing of the solvents. However, this method results in low concentrations of SLNs [135].3.Solvent Injection Method: The oil phase containing lipids and drugs is rapidly injected into an aqueous phase, which leads to direct droplet formation and SLN stabilization. However, this method requires exact control during the injection [136].Non-Solvent-based methods:1.High-Pressure Homogenization Method: includes hot and cold techniques. The hot method homogenizes a coarse emulsion [137]. While the cold method processes lipid micro particles in cold solutions [138]. The method’s limitations are related to temperature sensitivity and complexity [139].2.High-Speed Stirring and Ultra-Sonication Methods: This method depends on high-temperature mixing, mechanical stirring and cooling to form SLNs [140]. This method is cost-effective but exposes medications to high temperatures [141].3.The microemulsion method creates microemulsion by adding heated lipids and drugs to a surfactant solution, which is then diluted to yield SLNs [142]. However, this method involves a large amount of surfactant [141].4.The phase inversion temperature method uses temperature-dependent surfactants to create emulsions by heating above a specific temperature and then cooling to produce SLNs. The downside of this method is that it can lead to low stability of the molecules [143].5.The membrane contactor method uses a specific membrane to create lipid droplets in an aqueous phase, producing SLNs by cooling. This method requires advanced equipment [144,145].Other methods:1.The supercritical fluid-based method uses supercritical fluids like CO_2_ to facilitate SLN production but requires expensive fluids [146].2.The double emulsion method forms a water/oil/water double emulsion. This method is effective for hydrophilic drugs but is prone to high drug loss and large sizes [146].

Solid lipid nanoparticles have gained popularity as drug delivery methods for various cancer treatments. In liver cancer, SLNs have been used for targeted delivery of antisense oligonucleotides to liver endothelial cells, enhancing treatment efficacy for hepatocellular carcinoma. In breast cancer, SLNs not only improve drug delivery but also overcome multidrug resistance through passive targeting and surface modifications that extend circulation time. For colorectal cancer, SLNs can carry both hydrophilic and lipophilic drugs such as Oxaliplatin, enhancing their delivery to colon tumors, with a flexible composition that allows customization based on specific drug and administration needs [147].

Multiple drug release models from SLNs show how different drugs are released. Different drugs have significantly different release rates [148]. For example, for Tetracaine, high-pressure homogenization during preparation causes the drug to initially dissolve in the water phase. Once cooled, the lipids begin to crystallize, allowing the drug to redistribute into the lipid phase, which can potentially lead to rapid release from the outer shell. In the case of Etomidate, a more uniform solid solution model was observed where the drug is evenly distributed within the nanoparticle, resulting in more predictable release. Prednisolone, on the other hand, offers a different story, as it releases slowly over more than 5 weeks. This slow release is attributed to a drug-rich core surrounded by a nearly drug-free lipid shell, which makes it hard for the drug to diffuse out [149]. When paclitaxel is encapsulated in solid lipid nanoparticles (SLNs), only 0.1 percent is released after two hours, indicating a sustained release profile [150]. Similar release patterns have been noted for Doxorubicin and Idarubicin as well [151]. In contrast, for Cyclosporin A, less than 4 percent is released within 2 h, whereas in solution form, 60 percent is released in the same time frame [152].

A review by Durgaramani et al. [153] highlighted the advancements of SLNs in cancer therapy, focusing on their thermosensitive properties and potential as drug delivery systems. SLNs are submicron colloidal carriers made of biocompatible lipids and surfactants, capable of encapsulating both hydrophilic and hydrophobic drugs. Their thermosensitive behavior enables temperature-triggered drug release, as evidenced by the rapid release of 5-fluorouracil (>90%) at 39 °C, compared to 22–34% at 37 °C. SLNs demonstrate enhanced bioavailability, cellular uptake, and targeted delivery via both passive mechanisms (the EPR effect) and active targeting (surface modifications, such as folic acid or hyaluronic acid). Examples include paclitaxel- and doxorubicin-loaded SLNs for breast and lung cancer, achieving higher efficacy and reduced toxicity. SLNs have also shown success in delivering curcumin for breast cancer and erlotinib for lung cancer. Despite their advantages, challenges like low drug loading and stability remain. Future developments focus on surface modifications and a better understanding of SLN behavior in vivo to enhance their effectiveness in cancer therapy.

Fakhar ud Din et al. [154] conducted a study that highlighted a novel method for delivering the chemotherapy drug irinotecan using a double-reverse thermosensitive hydrogel (DRTH). This system utilizes SLNs in conjunction with a temperature-sensitive hydrogel to facilitate localized and controlled drug release. To produce the SLNs, a lipid mixture was melted, mixed with irinotecan, and then combined with a surfactant solution. Cooling the mixture caused the lipid to become more solid, thereby facilitating the encapsulation of the drug inside nanoparticles. These SLNs were then dispersed into a hydrogel made from Poloxamer 407 and Poloxamer 188. The hydrogel tends to remain in liquid form even at lower temperatures, allowing for easy handling and administration, but transforms into a gel at body temperature to enable drug release. At 25 °C (room temperature), the hydrogel remained in a liquid form, with only a small amount of drug release measured, ensuring it could be stored and transported efficiently. At 37 °C (body temperature), the hydrogel became more gel-like, causing a controlled and steady release of irinotecan over time. Furthermore, at elevated temperatures (above body temperature), the lipids within the SLNs lost stability, allowing a faster release of the drug. This property is particularly promising for treatments that involve hyperthermia, where localized heating could promote easier drug delivery to cancerous cells. Overall, this combination of SLNs and thermosensitive hydrogel provides a smart and effective method for targeting chemotherapy, while also minimizing side effects.

Another study by Sai et al. [155] incorporated the thermoresponsive polymer poly(N-vinylcaprolactam) (PVCL) with the developed SLNs for the controlled and targeted delivery of the hydrophilic drug gemcitabine hydrochloride. The SLNs were prepared using a double emulsion solvent evaporation method, with PVCL integrated into the core to enhance drug encapsulation and minimize burst release through hydrophilic interactions with the drug. Standard SLNs showed a 49.37% burst release in the first two hours, but PVCL-containing SLNs considerably decreased this to 33.4%, allowing for more controlled and sustained release, according to drug release experiments that showed a biphasic release profile at 37 °C. Drug release was somewhat decreased at 25 °C due to decreased diffusion; however, because PVCL was protected by the inner aqueous phase, its thermoresponsiveness was constrained, resulting in constant release profiles at both temperatures.

In order to enhance the skin distribution of tacrolimus, a medication used to treat atopic dermatitis that is poorly soluble in water, a study by Kang et al. [156] concentrated on thermosensitive SLNs. To modify their melting points, the SLNs were prepared using two distinct surfactants: SLN-1, which contained Poloxamer 188 and had a melting point of 48 °C, and SLN-2, which contained Brij 58 (polyoxyethylene (20) cetyl ether) and had a melting point of 36 °C. In a 24 h period, SLN-1 and SLN-2 both released 29% and 37.8% of tacrolimus at body temperature (37 °C), respectively, indicating temperature-triggered release. In comparison to SLN-2 and the brand-name ointment Protopic, SLN-1 was able to penetrate deeper layers of the skin (up to 450 μm). Studies conducted in vivo and ex vivo verified that Tacrolimus was better retained in dermal layers with SLN-1 and had little systemic absorption. At the dermal temperature (37 °C), the solid lipid nanoparticles melted and released tacrolimus, but they stayed stable at the outer skin temperature (32 °C). Tests for skin irritation showed that SLN-1 was safer since it caused less erythema. The promise of SLN-1 for targeted medication administration in skin disorders was highlighted by its overall regulated release, deeper penetration, and decreased adverse effects.

Mubashar et al. [157] used thermosensitive solid lipid nanoparticles (SLNs), which can release pharmaceuticals at hyperthermic temperatures (~39 °C), as a platform for targeted medication delivery in cancer therapy. Lauric acid was used as the solid lipid and oleic or linoleic acid as the liquid lipid, together with surfactants such as Brij 58 and Span 80, to create SLNs using 5-fluorouracil (5-FU) as a model medication. The SLNs remained stable at 37 °C, the usual physiological temperature, and released less than 28% of 5-FU over the course of five hours. Nevertheless, the SLNs underwent a solid–liquid phase change at 39 °C (a tumor-mimicking hyperthermic temperature), releasing more than 90% of the medication in the same amount of time. Electrochemical detection techniques and dissolving techniques were used to verify this thermoresponsive release. Compared to the limited cytotoxicity observed at 37 °C, cytotoxicity experiments demonstrated a significantly increased efficiency against breast cancer cells (MDA-MB-231) at 39 °C, resulting in a decrease in cell viability to 72–78%. At lower temperatures, the SLNs demonstrated a sustained release mechanism and were biocompatible with healthy human cells. These findings demonstrate the promise of thermosensitive SLNs as an easy, affordable, and effective medication delivery method for cancer treatment brought on by hyperthermia. Table 6 summarizes the most recent in vitro studies that have used thermosensitive solid lipid nanoparticles for cancer treatment. The table includes the components, payloads, cancer cell lines used in these studies, and the main findings related to drug release.

## 4. Preclinical and Clinical Applications of Thermosensitive Nanoparticles for Cancer Therapy

A modern method in cancer treatment is the use of stimuli-responsive drug delivery systems, which allow for localized, regulated, and effective drug release in response to particular triggers such as temperature, ultrasound, or tumor microenvironment (TME) characteristics. Numerous nanoplatforms, such as dendrimer nanodots [161], Janus micelles [162], thermosensitive hydrogels [163], liposomes, and PLGA-based nanoparticles [164], showed impressive therapeutic results in the six trials that were examined. In a 4T1 breast tumor model, for example, ultrasmall dendrimer nanodots activated by near-infrared light produced robust immunogenic cell death (ICD) and systemic CD8^+^ T-cell responses in addition to achieving 99% primary tumor shrinkage and 98.5% suppression of lung metastases [161]. In a GBM model, Janus micelles contained in a thermosensitive hydrogel allowed gemcitabine and R848 to be released sequentially, reducing tumor volume by 89.5% and strongly upregulating immune markers such as CD80, IFN-γ, and TNF-α [162].

Mitoxantrone-loaded thermosensitive liposomes, which were activated by modest hyperthermia (41 °C), demonstrated substantial tumor suppression and quick drug release in prostate cancers [165]. In triple-negative breast cancer models, PLGA nanoparticles in conjunction with perfluoropentane and external ultrasound demonstrated strong tumor volume reduction, reduced angiogenesis, and increased apoptosis [166]. In a different investigation, siRNA-loaded thermosensitive hydrogels continued tumor suppression for 56 days and produced sustained release over 7 days in ovarian cancer without causing systemic damage [163]. These results demonstrate the adaptability and potency of thermo- and stimuli-responsive mechanisms in both destroying tumor cells and stimulating the immune system for sustained defense.

High tumor selectivity, reduced off-target toxicity, improved immune activation (particularly in ICD-driven devices), and prolonged, adjustable release over time are some of these systems’ main advantages. Single-dose treatments with long-lasting effects are possible on many platforms [161,163]. Nevertheless, there are still restrictions: certain systems depend on external activation (such as laser or ultrasound), which could restrict their clinical scalability or accessibility. Others, such as intricate micelle–hydrogel systems, can be difficult to produce in large quantities or have poor penetration in deep tumor tissues. Additionally, not much research has been performed to assess long-term safety or possible immunological overactivation [162,166]. Notwithstanding these limitations, by tackling the fundamental issues of drug resistance, toxicity, and lack of tumor selectivity, these cutting-edge delivery techniques present a bright future in oncology. These technologies are laying the groundwork for more individualized and efficient cancer treatments by combining immune regulation with localized delivery. Table 7 summarizes some of the most recent preclinical and clinical studies of thermosensitive nanoparticles for precise cancer treatment.

Thermosensitive nanoparticles, particularly liposomes, have demonstrated significant clinical advancements in cancer therapy. Numerous clinical trials have explored the use of lyso-thermosensitive liposomal doxorubicin in combination with hyperthermia techniques, such as radiofrequency ablation (RFA) and MRI-guided high-intensity focused ultrasound (MR-HIFU). In contrast, other thermosensitive nanoplatforms—such as micelles, hydrogels, dendrimers, and solid lipid nanoparticles—are still in the preclinical phase, with no current clinical trials focused on hyperthermia-triggered drug release in cancer patients. As a result, liposomes remain the most clinically advanced thermosensitive carriers in oncology. The following discussion will provide examples of clinical trials involving thermosensitive liposomes.

Tak et al. [167] conducted a randomized trial to evaluate whether adding LTLD improves RFA for hepatocellular carcinoma (HCC) tumors (3–7 cm). Among 701 patients, no significant improvement in progression-free or overall survival was seen with LTLD + RFA compared to RFA alone. However, a subgroup with solitary tumors and ≥45 min of RFA showed significantly better overall survival. The combination was safe, with manageable side effects. Lyon et al. [33] conducted an open-label, Phase 1 trial in the UK to assess the safety and effectiveness of combining LTLD with focused ultrasound hyperthermia in patients with unresectable or non-ablatable liver tumors. Ten patients received a single LTLD infusion followed by focused ultrasound targeting one tumor. The treatment achieved the primary goal: at least double the intratumoural doxorubicin concentration in 70% of patients, with an average 3.7-fold increase. Side effects included expected grade 4 neutropenia and one case of mild confusion. The approach was found to be safe, feasible, and capable of significantly enhancing local drug delivery in difficult-to-treat liver tumors. T.M. Zagar et al. [168] reported results from two Phase I trials evaluating LTLD combined with mild local hyperthermia (MLHT) for treating unresectable chest wall recurrences (CWR) of breast cancer in heavily pretreated patients. Twenty-nine patients, many with prior extensive chemotherapy, radiotherapy, and metastases, received up to six LTLD cycles followed by 1 h MLHT at 40–42 °C. Treatment was generally well tolerated, with reversible grade 3–4 neutropenia (24%) and leukopenia (14%). An overall local response rate of 48% was observed, including 17% complete and 31% partial responses. The study demonstrated that LTLD plus MLHT is safe and can provide meaningful tumor responses in this challenging patient group. Another Phase I trial by de Maar JS et al. [169] investigates the safety, feasibility, and tolerability of combining LTLD with MR-HIFU-induced hyperthermia in breast cancer patients. The goal is to replace conventional doxorubicin in the AC regimen with LTLD and focused heat to enhance local tumor response without increasing systemic toxicity. Patients received up to six cycles, with MR thermometry guiding hyperthermia at 40–42 °C for 30 min. If hyperthermia was ineffective or a complete radiological response occurred early, standard treatment resumed. No tissue biopsies were taken; local efficacy was assessed radiologically. The treatment was found to be safe and feasible, supporting further investigation into larger studies. If validated, this approach could improve neoadjuvant therapy and benefit other doxorubicin-sensitive cancers. Table 8 summarizes the completed clinical studies on thermosensitive liposomal drug delivery for cancer therapy.

## 5. Challenges and Future Direction

Although smart thermosensitive nanocarriers show great promise for precision cancer therapy, their clinical use is currently limited by various scientific, material, regulatory, and operational challenges. It is essential to address these limitations to fully realize their therapeutic potential and facilitate widespread adoption in clinical settings.

The most prominent clinical example of thermosensitive nanocarriers is ThermoDox^®^, a lysolipid-based thermosensitive liposome encapsulating doxorubicin. ThermoDox^®^ is the only thermosensitive liposome to reach Phase III clinical evaluation. Despite promising preclinical data, ThermoDox^®^ failed to meet its primary endpoints in large-scale clinical trials, such as the HEAT and OPTIMA studies for hepatocellular carcinoma [33]. For example, Dou et al. [170] note that clinical trial design challenges and data gaps have limited its use. These failures were due to inadequate thermal dosing, uneven heating in tumor tissues, and variations in tumor blood vessels, leading to inconsistent drug release and therapeutic outcomes. Additionally, challenges in integrating external hyperthermia techniques—like HIFU, microwave heating, or radiofrequency ablation—into standardized protocols limit the reproducibility and scalability of these treatments [171]. Moreover, many thermosensitive nanoparticle systems are still limited to preclinical settings due to the lack of a standardized protocol for hyperthermia. Additionally, controlling localized temperatures in deep or heterogeneous tumors poses significant challenges, and real-time temperature monitoring remains difficult, even with the use of MR thermometry.

Micelles are easily fabricated and ideal for encapsulating hydrophobic drugs. However, their stability in vivo is poor due to dissociation below their CMC, resulting in premature drug leakage and poor systemic retention [172]. The clinical translation of thermosensitive micelles for therapeutic applications faces several regulatory challenges. Key factors include biocompatibility, manufacturing scalability, and regulatory compliance, all of which are vital for advancing micellar drug delivery systems. Thorough evaluation of the potential toxicity is essential for ensuring patient safety, as regulatory bodies require comprehensive toxicity studies before approval [173]. Moreover, the risk of immune responses against micelles requires strict testing to confirm biocompatibility [174]. Scaling up production while maintaining quality poses a challenge, as current manufacturing methods may not support large-scale output. Economic feasibility is critical for commercial viability [175]. Navigating the evolving regulatory guidelines for nanomedicines is crucial, as delays can hinder clinical translation [176]. Additionally, securing patents and addressing intellectual property rights can complicate development and impact investment in micelle technology [175].

Hydrogels are attractive for localized and sustained release due to their high water content and tunable crosslinking. However, they are often not suitable for systemic administration, and their large size and dense network can limit heat penetration and drug diffusion [177]. Thermosensitive hydrogels have potential in therapeutic applications due to their sol–gel transition properties, which enable targeted drug delivery. However, their clinical translation faces several regulatory challenges, including formulation issues like poor drug loading, difficulties in controlling drug release kinetics [178], and ensuring biocompatibility and stability [179]. The lack of preclinical models for certain conditions, such as Carpal Tunnel Syndrome, hampers clinical validation [180]. Very few thermoresponsive hydrogel-based drugs have entered clinical trials, indicating a need for rigorous testing to confirm safety and efficacy. The regulatory pathway is complex, with minimal FDA-approved products, particularly in cancer therapy [178]. Addressing formulation challenges, such as low response time and storage issues, is crucial for meeting regulatory standards [179].

Dendrimers offer high functionalization and loading capacity due to their branched architecture. Still, their synthesis is expensive and complex, and their cationic surfaces may induce cytotoxicity, hemolysis, and immune responses at higher concentrations [181]. The clinical translation of thermosensitive dendrimers faces several regulatory challenges, particularly regarding synthesis, safety, and compliance. Key points include the need for scalable and reproducible synthesis methods, the importance of achieving optimal surface multifunctionality without sacrificing drug loading capacity, and understanding the toxicity and clearance mechanisms for long-term safety [182]. Consistent quality attributes across batches are essential, requiring robust analytical methods. Additionally, the lack of clear regulatory guidance for nanomedicines poses hurdles, making early collaboration with regulatory agencies vital for aligning development with expectations [183].

Liposomes, especially thermosensitive types like ThermoDox^®^, are the most clinically advanced, but they are limited by drug leakage, batch-to-batch variability, and the requirement for precise thermal triggering [184]. The clinical translation of thermosensitive liposomes faces several regulatory challenges due to their complex formulations, which require thorough evaluation and adaptation of existing guidelines. These liposomal products necessitate extensive characterization of their physical and chemical properties, impacting the regulatory review process compared to traditional drugs. While the FDA and EMA have developed specific guidelines, these guidelines are still evolving and may lag behind rapid advancements in liposomal technology, potentially leading to compliance gaps [185]. Comprehensive preclinical and clinical studies are crucial for evaluating the safety and efficacy of thermosensitive liposomes, including their immunogenicity and potential adverse effects. Consistent production across batches is critical to ensuring therapeutic outcomes and regulatory approval. Furthermore, the innovative nature of these liposomes raises ethical considerations regarding patient safety and informed consent [186].

SLNs exhibit good biocompatibility and physical stability, but their rigid crystalline structure can hinder thermal responsiveness and limit drug loading. The emergence of nanostructured lipid carriers (NLCs) aims to overcome some of these drawbacks [139]. The clinical translation of thermosensitive SLNs for therapeutic applications faces several regulatory challenges. These include formulation complexities, safety assessments, and compliance with regulatory frameworks, which are essential for moving SLNs from research to clinical use. First, a precise formulation is necessary to ensure the stability and efficacy of SLNs, which complicates the manufacturing process. Additionally, maintaining stability during storage and application is crucial, as instability can reduce therapeutic effectiveness. Second, obtaining regulatory approval presents hurdles, as SLNs must meet strict safety and efficacy standards. Comprehensive safety evaluations are needed to assess biocompatibility, which can prolong the approval process. Lastly, navigating intellectual property rights can complicate commercialization and impact investment. Economic viability compared to existing therapies is also a key consideration for regulatory bodies [175]. Despite these challenges, opportunities exist for innovation in formulation and regulatory strategies, which can enhance the clinical translation of this promising drug delivery system.

To address these multifaceted challenges, several strategic directions should be pursued. First, combining temperature responsiveness with other triggers, such as pH, enzymes, or light, can enhance site specificity and improve release profiles. Dual- or multi-responsive nanocarriers are more likely to overcome tumor heterogeneity and microenvironmental barriers [187]. Furthermore, integration of MR-guided HIFU, photoacoustic imaging, and thermosensors can enable real-time thermal monitoring and personalized control of hyperthermia, improving the spatial accuracy of triggered drug release [188]. In addition, machine learning and computational models can assist in predicting nanoparticle behavior, optimizing design parameters, and personalizing treatment protocols based on patient-specific tumor profiles [189].

## 6. Conclusions

As cancer incidence continues to rise across the globe, there is an urgent need for more effective, safer, and targeted therapeutic strategies. Traditional cancer treatments, while often effective to some extent, are frequently accompanied by severe side effects due to their non-specific nature and systemic toxicity. In response, researchers have increasingly turned to innovative drug delivery systems that can enhance therapeutic efficacy while reducing harm to healthy tissues. Among these, stimuli-responsive drug delivery systems (SDDSs) have emerged as a particularly promising class of technologies. Of special interest are those SDDSs that are responsive to temperature changes, offering site-specific, controlled drug release when exposed to mild hyperthermia.

Recent advances in nanotechnology have played a crucial role in this development, enabling the engineering of smart nanocarriers that can navigate complex biological environments and respond to external or internal stimuli. These thermosensitive nanocarriers can accumulate at tumor sites via the enhanced permeability and retention (EPR) effect, releasing their payload precisely when and where needed, often triggered by externally applied heat. The synergy between hyperthermia and thermosensitive delivery systems thus represents a powerful approach in cancer therapy, offering improved selectivity and reduced systemic exposure.

This review has highlighted the growing role of nanoparticles in the design of smart drug delivery systems. Natural carriers, such as lipids, polymers, and proteins, provide a biocompatible and biodegradable platform for developing thermoresponsive systems. Their intrinsic properties, including ease of functionalization, low immunogenicity, and tunable release profiles, make them ideal candidates for next-generation cancer therapeutics. At the same time, this review addresses the limitations and challenges associated with these carriers, including stability, scalability, and variability in biological response.

Looking ahead, future research should prioritize the clinical translation of these promising technologies. This includes a thorough investigation of their pharmacokinetics, biodistribution, toxicity profiles, and therapeutic efficacy in relevant animal models and human trials. Additionally, optimizing the combination of hyperthermia parameters with nanoparticle design will be crucial to maximize therapeutic benefits. Ultimately, with continued interdisciplinary collaboration and technological innovation, thermosensitive SDDSs have the potential to revolutionize cancer treatment by offering highly effective, targeted, and patient-friendly alternatives to conventional therapies.

## Figures and Tables

**Figure 1 ijms-26-07322-f001:**
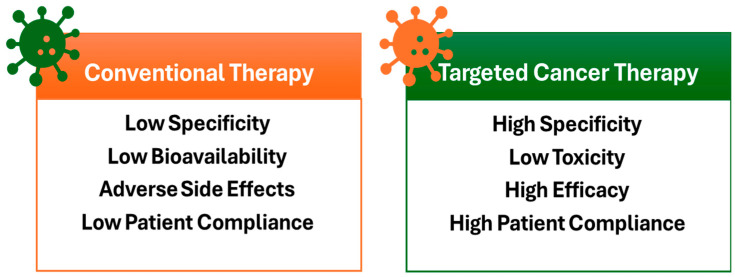
Characteristics of various drug delivery methods.

**Figure 2 ijms-26-07322-f002:**
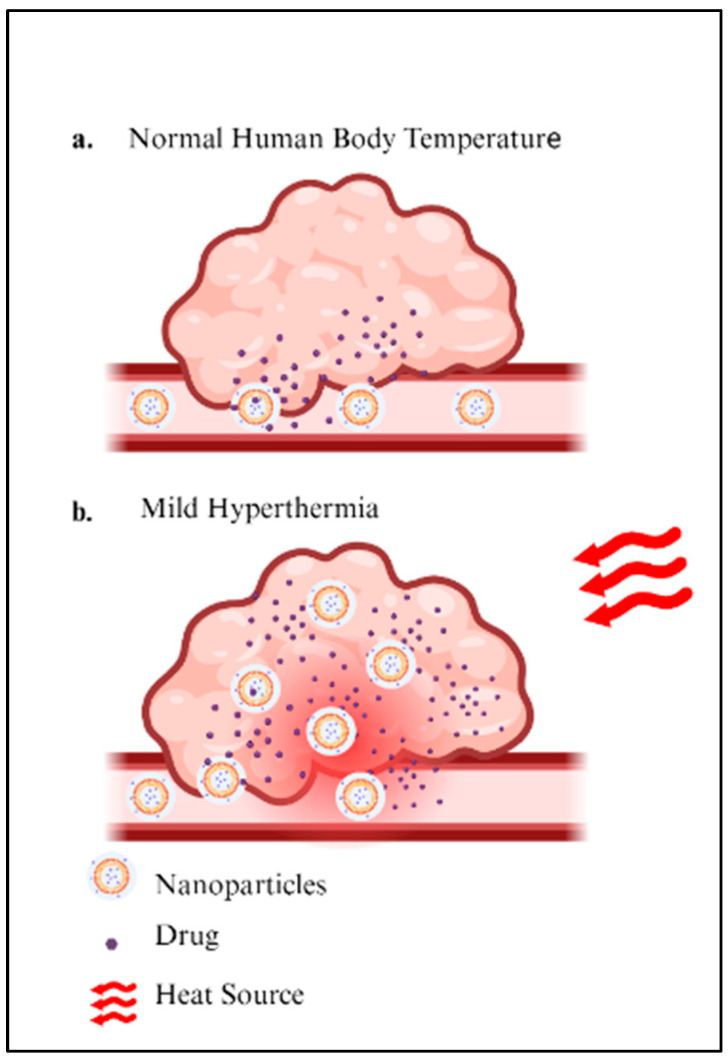
The effect of mild hyperthermia on temperature-sensitive nanoparticles (Created in Biorender. Atena, Y. (2025) https://BioRender.com).

**Figure 3 ijms-26-07322-f003:**
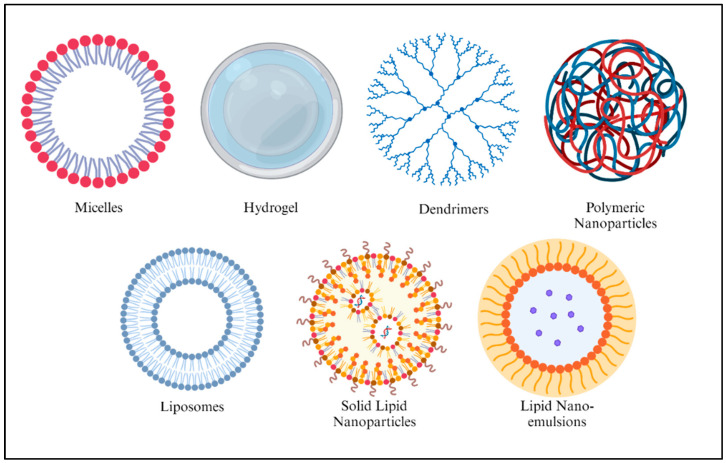
Nanocarriers that are commonly used with thermal stimuli (Created in Biorender. Atena, Y. (2025) https://BioRender.com).

**Figure 4 ijms-26-07322-f004:**
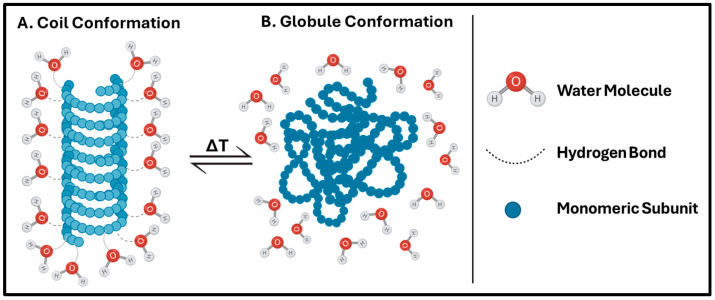
Temperature-dependent reversible phase transition of the thermosensitive polymers (Created in Biorender. Atena, Y. (2025) https://BioRender.com).

**Figure 5 ijms-26-07322-f005:**
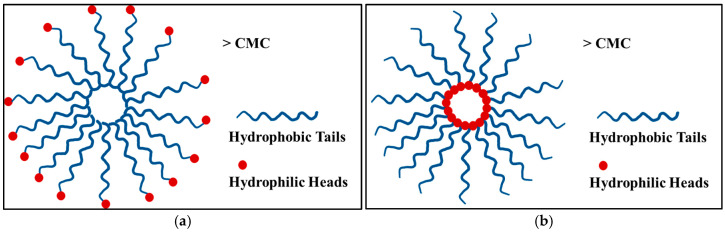
Self-assembly of polymer into (**a**) spherical and (**b**) inverse micelles.

**Figure 6 ijms-26-07322-f006:**
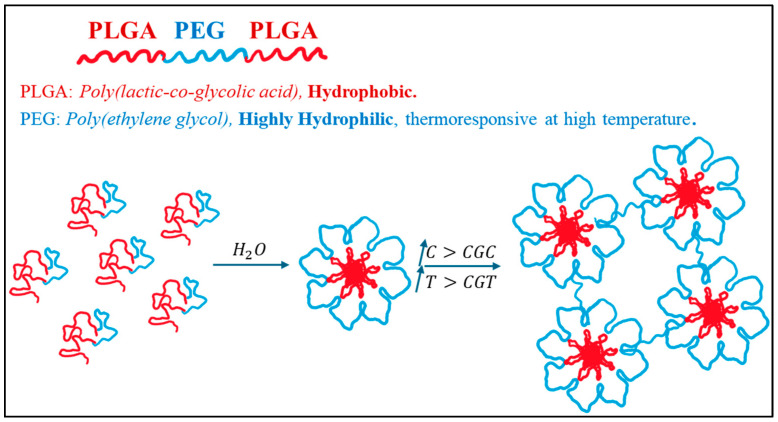
The PLGA-b-PEG-b-PLGA polymer (BAB pattern). As temperature increases toward the CGT (↑T>CGT) in highly concentrated solutions (↑C>CGC), hydrophobic interactions increase, leading to strong micellar aggregation, loss of flowability, and gelation.

**Figure 7 ijms-26-07322-f007:**
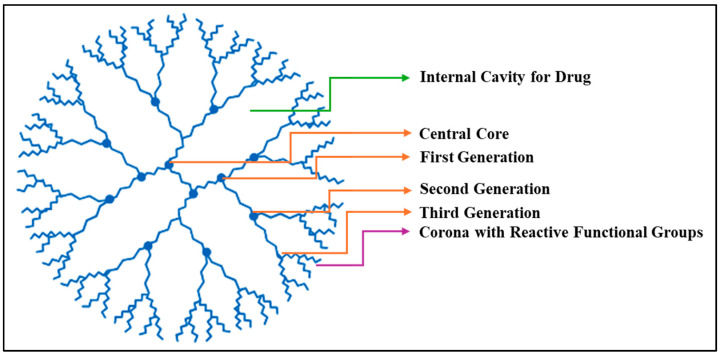
Schematic representation of the molecular structure of dendrimers illustrating their highly branched tree-like architecture.

**Figure 8 ijms-26-07322-f008:**
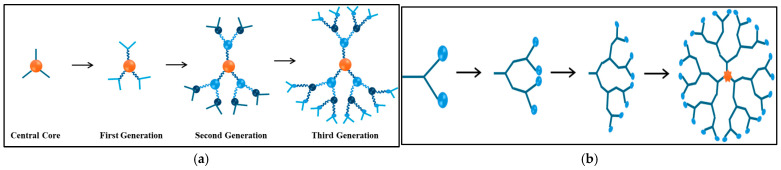
A schematic representation of (**a**) the divergent and (**b**) the convergent synthesis approach.

**Figure 9 ijms-26-07322-f009:**
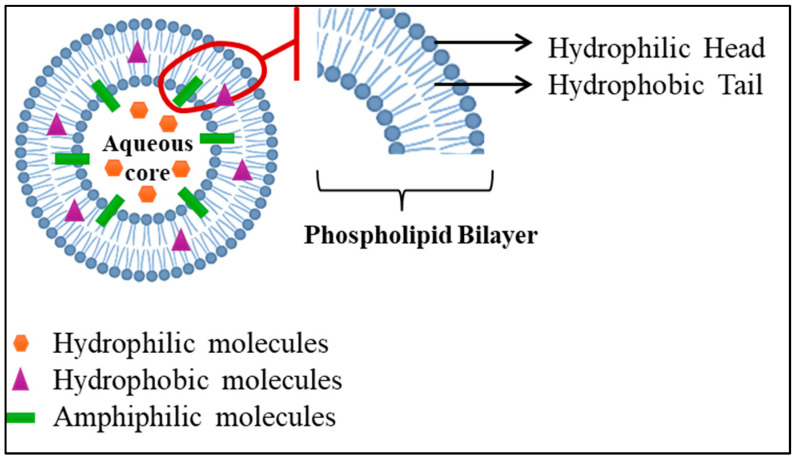
The general structure of a liposome.

**Figure 10 ijms-26-07322-f010:**
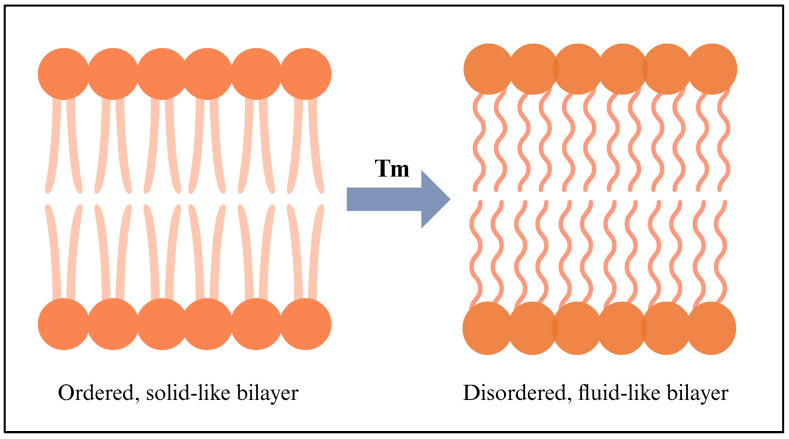
Lipid bilayer phase transition (Created in BioRender. Atena, Y. (2025) https://BioRender.com).

**Figure 11 ijms-26-07322-f011:**
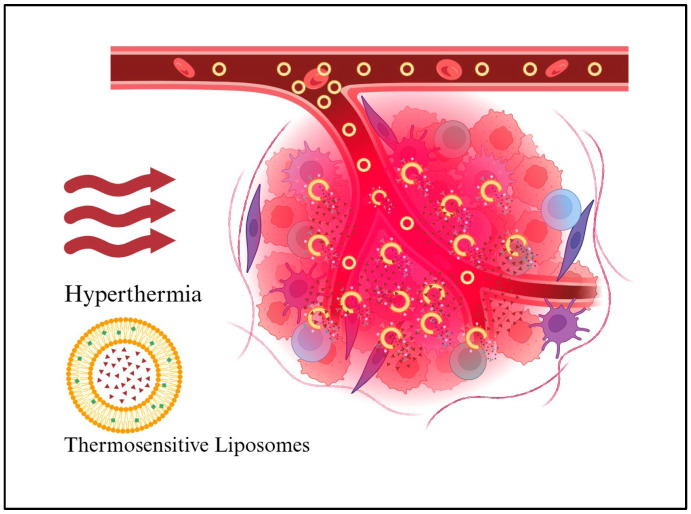
Thermosensitive liposomes function when hyperthermia occurs. The lipid shell is shown in yellow, and the drug is represented by red triangles. (Created in BioRender. Atena, Y. (2025) https://BioRender.com).

**Figure 12 ijms-26-07322-f012:**
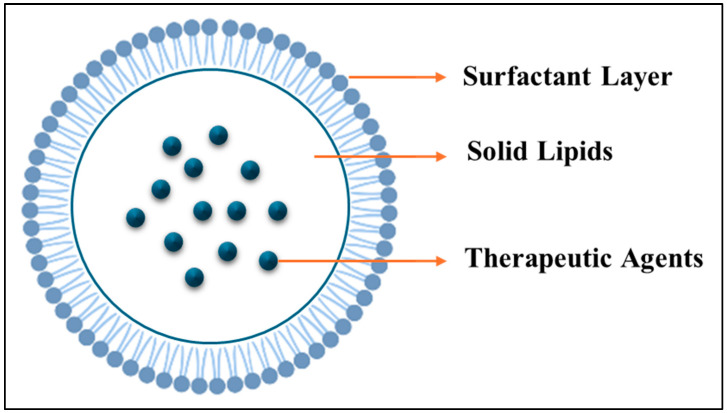
A schematic representation of the molecular structure of a solid lipid nanoparticle.

**Table 1 ijms-26-07322-t001:** Temperature-responsive polymers (adapted with changes from [50,51,52,53]).

Polymer Name	Applications	Polymerization Methods	Properties
Poly(N-isopropyl acrylamide) (PNIPAAm)	Drug and gene delivery, bioseparation, and cell culture	Free radical polymerization, living radical polymerization	Temperature-responsive when mixed with water and organic solvents. Solvent–polymer interactions, coexistence of LCST and UCST, concentration-dependent behavior, LCST stability, amphiphilic end groups influence the LCST, LCST increases with ionic surfactants, polyelectrolyte behavior, entropy of counterions, the LCST of PNIPAM is influenced by the type of salt present in the solution.
Multi-block copolymers (ABA, BAB, etc.)	Drug delivery, tissue engineering, and injectable gels	Atom transfer radical polymerization (ATRP), reversible addition-fragmentation chain-transfer (RAFT), oxyanionic	Self-assemble and stabilize in solution. It can promote microphase separation and facilitate the microphase separation within gels (either physical or covalently linked gels).
Star-shaped poly (2-alkyl-2- oxazolines)	Sensors, rheological additives, and multiple biological applications, including drug delivery	Divergent (“core-first”) and the convergent (“arm-first”)	Self-association in solution depends on solvent composition, temperature, and the microscopy analysis.
Poly(N-vinylcaprolactam) (PVCL)	Biosensing, controlled drug delivery, and stimuli-dependent targeting	RAFT, ring-opening polymerization (ROP) combined with ATRP, and cobalt-mediated radical polymerization (CMRP)	The phase transition in solution occurs above 35 °C upon heating. The temperature response is dependent on its molecular weight, polymer concentration, polymer chemical composition, cosolvent, ionic strength, and surfactants.

**Table 2 ijms-26-07322-t002:** A summary of the most recent in vitro studies relevant to thermosensitive micelles in cancer therapy.

Components	Payload	Cancer Cell Line	Synthesize Method	Temperature-Triggered Release	Ref.
PVS-b-PNVCL PVL-b-PNVCL	DOX	HEK 293 T HeLa	RAFT	PVS18-b-PNVCL35 released 46–69% within 24 h and 53–90% within 72 h at different temperatures (25–37 °C). DOX release from PVS18-b-PNVCL95 was 78% within 72 h. Free DOX was released at 64.2% in 2 h and approximately 90% in 10 h.	[63]
Alg-g-PNIPAAm	Dipyridamole (DP)	HEK293T HeLa HCT116	RAFT	The transmittance value decreased to 50% of its initial value at 32 °C.	[64]
SP94-PEG-p(AAm-co-AN)/LEN/IR-1061-AcD, SPLI	Lenvatinib (LEN)	H22 AML-12 Luc-H22	Free Radical Polymerization	H22 cells treated with Nile red co-loaded micelles showed increased fluorescence under NIR laser, indicating micelle destruction and enhanced cellular release of Nile red. SPLI/Nile red exhibited higher fluorescence intensity, demonstrating better uptake by the cells.	[65]
PME3CL-b-PBnCL PME3CL-b-PPhCL PME3CL-b-PEtOPhCL	DOX	MDA-MB-231	ROP	The initial release rate of PME3CL-b-PEtOPhCL micelles was the highest. (60% after 8 h). A polymer with a lower LCST shows a faster release.	[66]
Chit5-SA-20 Chit5-OA-20 Chit5-MUA-20 Chit5-LA-20	DOX	A549 HEK293T	Not specified	The rate of Dox release increases by 1.5–3 times with temperature increases from 25 °C to 42 °C. The intensity of all peaks, especially N–H and O–H fluctuations, increases in the temperature range of 22–45 °C.	[67]
PNIPAM–PEG–PNIPAM	Zinc Protoporphyrin (ZnPP)	PC3	RAFT	The ZnPP release was higher at 27 °C than at 37 °C. The ZnPP release was 48.2% and 11.35% within 0.5 h at 27 °C and 37 °C, respectively. The ZnPP release was 90.4% and 59.6% within 48 h and 93.1% and 62.45% within 72 h at 27 and 37 °C, respectively.	[68]
PNOG-b-PNAG-b-PNMG	Ciprofoxacin (Cip)	NCTC clone 929 (L-929)	ROP	The release rate of Cip without polymer was 100% in 3 h. The drug release rate with polymer wrapping was 56% at 20 h.	[69]

**Table 3 ijms-26-07322-t003:** A summary of the most recent in vitro and in vivo studies relevant to thermosensitive hydrogels in cancer therapy.

Components	Payload	Cancer Cell Line/Animal Model	Synthesize Method	Release Main Findings	Ref.
PDLLA-PEG-PDLLA, PLEL	Gemcitabine (GEM) Cytosine-phosphate-guanine oligonucleotide (CpG-ODN)	MB49 C57BL6	Ring-Opening Copolymerization	The GEM release rates at 24 h and 10 d were 57.7% and 90.8%, respectively. The CpG release at 24 h and 96 h was 60.8% and 94.2%, respectively. Both were released entirely from PLEL on day 7.	[77]
mPEG2k-PAla32-block-PAsp5	RMPs@Mn^2+^	B16-F10 H22 C57BL/6 BALB/c	Not specified	At pH 6.8, Mn^2+^ release reached 82.6%.At pH 7.4, Mn^2+^ release reached 60.3%.The free Mn^2+^ rapidly accumulated in the liver only 15 min after injection. Gel@Mn^2+^ slowly released Mn^2+^.	[78]
Cy7-Cell@hydrogel (CT26-loaded Pluronic^®^ F-127/gelatin)	Cyanine dyes (Cy7)	CT26	Physical Crosslinking	(40 μg/mL) Cy7 exhibited a significant temperature increase (45 °C) at a laser power density of 0.9 W/cm^2^. The Cy7-Cell@hydrogel system can efficiently convert light into heat energy for up to 80 min.	[79]
PLGA-PEG-PLGA	Epidermal growth factor (EGF)	NCG HeLa	Ring-opening Polymerization	The tumor grew significantly faster in the free EGF group than in the hydrogel-EGF group. The weight and volume of tumors from the control group were significantly larger than those in the hydrogel-EGF group. HeLa cells showed more substantial growth potential in free EGF than in hydrogel-EGF. Hydrogels’ sustained EGF release behavior could effectively inhibit tumor growth.	[80]
HTPM and AgNPs-gel C6/HTPM and AgNPs-gel	Halofuginone hydrobromide (HF) Ag+ Coumarin 6 (C6)	HUVEC MDA-MB-231 MDA-MB-231-luc Eph4-ev BALB/c	Physical Swelling Method	Compared with the 60% release of the HF loaded into the HTPM-gel, the release pattern of the HF loaded into the HTPM and AgNPs-gel was slow and constant over 24 h. Compared with the 40% release of the AgNPs loaded into the AgNPs-gel, the release pattern of the AgNPs-gel loaded into HTPM and AgNPs-gel was slow and constant over 12 h.	[81]
Halofuginone-loaded D-α-tocopherol polyethylene glycol 1000 succinate (TPGS) polymer micelles nano-thermosensitive hydrogels (HTPM-gel)	Halofuginone	CMT-U27Eph4-evMDCK	Physical Swelling Method	Within 24 h, under a pH 6.5 environment, the CRP of halofuginone from HTPM rose to 64.48%. The CRP of halofuginone from HTPM-gel was approximately 71.18% within 24 h.	[82]
FCDL and DSF-SE@G	Cu^2+^ Disulfiram (DSF) Glycyrrhizic acid-Cu (FCDL)	MGC-803 Balb/C	Reverse-phase Evaporation method	Cu^2+^ and DSF release in the FCDL and DSF-SE@G was 53% and 46%, respectively, which results in sustained drug release.	[83]
PDLLA-PEG-PDLLA (PLEL)	Cabazitaxel (CTX)	HCT-15 HCT-116 Balb/c-nu	Ring-opening Polymerization	At 48 h, the drug release rate of CTX/PLEL was approximately 11.4%. At 96 h, the release rate was only approximately 16.8%. The loading of PLEL significantly slowed the in vitro drug release of CTX.	[84]
DOPA-rGO@PC-gel	Dopamine-reduced graphene oxide (DOPA-rGO; photothermal nanoagent)	NHDF MCF-7	Dual-crosslinking Method	DOPA-rGO@PC-gel could produce a similar photothermal heating (ΔT ≈ 22 °C) at a considerably lower concentration and laser intensity (99.94 µg/mL of DOPA-rGO; 1.7 W/cm^2^), but it required a slightly longer irradiation time (10 min). These results further confirm the good photothermal capacity of the DOPA-rGO@PC-gel.	[85]

**Table 4 ijms-26-07322-t004:** A summary of the most recent in vitro and in vivo studies relevant to dendrimers in cancer therapy.

Components	Payload	Cancer Cell Line	Synthesize Method	Release Main Findings	Ref.
Fucosylated PAMAM Dendrimers	Chrysin (CRY)	A549 (Human Lung Cancer)	Fucosylation via Schiff’s base formation with fucose	At 37 °C, CRY showed sustained release in two media: pH 5 (endolysosomal conditions): 87.6% released after 24 h pH 7.4 (plasma conditions): 93.6% released after 24 h.	[103]
PSMA-Targeted Dendrimer Nanoplatform (PD-CTT1298-Cabo)	Cabozantinib (Cabo)	Prostate-specific membrane antigen (PSMA). PSMA-positive PC3-PIP cells and PSMA-negative PC3 cells. PSMA-positive PC3-PIP tumor xenograft mouse model	Strain-promoted azide–alkyne cycloaddition (SPAAC) chemistry	The release rates were monitored through incubation at 37 °C: Plasma-like conditions (pH 7.4, phosphate-buffered saline): Drug release occurred gradually under these neutral pH conditions. Simulates systemic circulation where slower release is favorable. Intratumoral conditions (pH 5.5, citrate buffer, containing esterase): Drug release was significantly faster under these acidic conditions. Mimics the tumor microenvironment with lower pH and enzyme activity, promoting rapid drug delivery.	[104]
Nanodiamonds (NDs) modified with PNIPAM Hyper-branched dendrimers	Docetaxel (DXL)	Breast cancer	Surface functionalization Polymer grafting Dendrimer attachment Drug adsorption	99.78% release at 45 °C (pH 5.6) in 6 h 69.58% release at 45 °C (pH 7.4) in 6 h 90.97% release in 20 min under NIR laser irradiation (808 nm, 1 W/cm^2^) Faster release due to polymer phase transition (LCST~32 °C).	[105]
PNIPAM/FeRh composite.	DOX	Not specified (general cancer model)	Solvent casting of PNIPAM on FeRh alloy; laser modification to create drug wells	Triggered by LCST (~32 °C) using magnetic field-induced cooling.	[101]
Amphiphilic dendrimer-like copolymer with a hydrophobic poly(styrene) core and hydrophilic PEO shell	Benzyl halide (benzyl chloride, benzyl bromide)	Not mentioned	An iterative divergent process involving anionic polymerization, hydrosilylation, chlorosilane coupling, and olefin cross-metathesis reaction	Release occurs due to the thermal responsiveness of the PEO segments forming the hydrophilic shell. LCST of PEO segments (~65 °C): Below LCST: PEO segments are hydrated, forming stable unimolecular micelles. Above LCST: PEO segments shrink, leading to aggregation and potential release of encapsulated hydrophobic molecules.	[100,101]

**Table 5 ijms-26-07322-t005:** A summary of the most recent in vitro and in vivo studies relevant to thermosensitive liposomes in cancer therapy.

Components	Payload	Cancer Cell Line/Animal Model	Synthesize Method	Release Main Findings	Ref.
Dox-Lipo/Gel/PGA/FeNP	DOX	MDA-MB-231-Luc	Solvothermal method and thin-film hydration method	DOX was released from the liposomes, while the liposomes were not released from the scaffold during incubation in the magnetic hyperthermia environment.	[124]
BDNF-LTSL-cRGD or BDNF-LTSL-HIT	Neurotrophin Brain-Derived Neurotrophic Factor (BDNF)	Mouse microvascular endothelial cell line (EOMA), Rat glomerular endothelial cell (rGECs), 7-week-old Sprague Dawley rats	Lipid film hydration and extrusion method	BDNF delivery and permeation across the co-culture filter were achieved by pre-heating BDNF-LTSL-cRGD at 42 °C for 30 min, which enabled the thermoresponsive release of the payload.	[125]
BI-FA-LP	Berberine (BBR), ICG (an FDA-approved photothermal agent (PTA))	4 T1 and RAW264.7 cell lines, Female BALB/C mice (4–5 weeks old)	pH gradient method	After irradiation with an 808 nm laser, 47.26 ± 0.53% of BBR and 56.39 ± 3.39% of ICG were released from BI-FA-LP. The thermosensitive drug release ability of BI-FA-LP provided the basis for sustained release and long circulation of liposomes in vivo.	[126]
PC + Mal-PEG-DSPE PC + Mal-PEG-DSPE + K3 PC + Mal-PEG-DSPE + 09JA PC + Mal-PEG-DSPE + K3 + 09JA Anti-HER-2-Fab modified liposome	DOX	The human stomach cancer cell N87, Balb/c nude mice, female, 4–6 weeks, ~20 g	Not specified	Compared to DOX, the suspension exhibited a rapid increase in drug release over time and eventually reached a plateau. No significant leakage of DOX from samples a, b, c, and d was observed after incubation for 60 min at 37 °C. The release of free DOX molecules was examined as a control group, which showed a complete cumulative release within 12 h. The releases of liposomes c and d were potentiated at 42 °C and exhibited thermo-triggered burst release of DOX.	[127]
BiNSs/Met/5-FU@TSL	5-fluorouracil (5-FU) and metformin (Met)	HT29 Healthy female Nod/scid mice (5–6 weeks old, 15–18 g)	Reverse phase evaporation method	In vitro drug release behavior of BiNSs/Met/5-FU@TSL is temperature-dependent. The release of the drugs is significantly accelerated at higher temperatures, resulting in a greater cumulative release within the specified time frame.	[128]
BioSi@NPs + TSLs + AuNPRs	Carboxy fluorescein (CF)	Not specified	Thin-film hydration method	A clear relationship was demonstrated between the release percentage and the applied power or exposure time, allowing for precise regulation of the fluorescent probe’s release kinetics.	[129]
LTLD + mEHT	DOX	The 4T1 triple-negative breast cancer (TNBC) cell line Six–eight-week-old female BALB/c mice	Not specified	LTLD releases 80% of DOX into the bloodstream in the heated tumor. mEHT did not enhance the DOX accumulation from PLD in the tumor. Strong proliferation inhibition was observed when DOX was combined with mEHT. LTLD caused stronger inhibition of proliferation than PLD.	[120]
ThermoDXR	DOX	4T1 cells female BALB/c mice at the age of 6–8 weeks	Thin film hydration method	The combination of the thermosensitive liposome formulation of doxorubicin and radiation therapy (RT), together with hyperthermia at reduced doses of both the drug and RT, not only induced a strong therapeutic effect but also reduced treatment-related toxicities.	[130]
NIR-II Photoexcited Lip(NiPTZ-GA)	Gambogic acid (GA)	Five- or six-week-old BALB/c mice 4T1 tumor–bearing mice	Not specified	Liposomes composed of GA-PEG, DSPE-PEG2000, and DPPC decomposed under 1064 nm laser irradiation, releasing NiPTZ. NiPTZ generated elevated temperatures for efficient NIR-II photoacoustic imaging (PAI) and photothermal therapy (PTT). Lip(NiPTZ-GA) demonstrated a high PTT conversion efficiency of 49%. The combination of PTT and chemotherapy achieved optimal therapeutic outcomes in vivo studies.	[131]

**Table 6 ijms-26-07322-t006:** A summary of the most recent studies relevant to SLNs in cancer therapy.

Components	Payload	Cancer Cell Line	Synthesize Method	Main Findings	Ref.
PAC-CUR-SLNs	Paclitaxel (PAC) Curcumin (CUR)	A549 (Lung Cancer) BALB/c Mice	High-Pressure Homogenization (HPH)	In vitro Release (37 °C): Sustained release observed: ~41% PAC, ~29% CUR released in 6 h. ~97–98% released over 96 h.	[158]
SLN-BBS-COPA	Lauric acid (BBS) β-Caryophyllene (COPA)	PC-3: Androgen-independent human prostate cancer cell line DU-145: Androgen-independent human prostate cancer cell line	Emulsification–ultrasonication	Refrigerated samples (8 °C) maintained stability over 60 days, while samples at 25 °C showed increased hydrodynamic diameter (HD) and PDI over time.	[159]
Lauric acid + Oleic/Linoleic acid	5-Fluorouracil	Human gingival fibroblast (HGF; PCS-201-108) Human breast adenocarcinoma cells (MDA-MB-231, HTB-26)	High-pressure homogenization	At 37 °C (normal physiological conditions): Sustained drug release observed. At 40–42 °C (mimicking tumor or hyperthermic conditions): Increased drug release, improving drug availability in targeted regions.	[157]
Superparamagnetic iron oxide NPs (SPIONs)	Dox	Not specified	Emulsification and solvent evaporation	At 37 °C: Minimal drug release under normal conditions. At 40–45 °C: Triggered release of doxorubicin due to temperature rise, synergistically enhanced by external magnetic fields and acidic tumor.	[160]

**Table 7 ijms-26-07322-t007:** A summary of preclinical studies of thermosensitive nanoparticles for cancer therapy.

Nanoparticle Type	Targeting Mechanism	Animal Model	Remarks	Ref.
Ultrasmall dendrimer nanodots	Near-infrared (NIR) laser activation for photodynamic therapy (PDT).	Orthotopic 4T1 breast tumor model in mice.	99% inhibition of primary tumor growth. Induced strong immunogenic cell death (ICD): CRT exposure, ATP/HMGB1 release, dendritic cell maturation. 98.5% suppression of lung metastasis Enhanced CD8^+^ T-cell infiltration and systemic antitumor immunity.	[161]
PLGA-core nanoparticle with DSPE-PEG modified by p-tosylethylenediamine (TSE-CEL/NP)	Triggered by external ultrasound.	C57BL/6 mice bearing B16F10 melanoma (primary, bilateral, and lung metastasis models).	Preferential accumulation in endoplasmic reticulum (ER) leads to strong ER stress (↑ p-IRE1α, ATF6, p-eIF2α; ↑ ATF4, GRP78, XBP1). Enhanced ICD markers: CRT exposure, HMGB1 and ATP release. Boosted immune response: ↑ immature and mature DCs, CD4^+^ and CD8^+^ T-cell infiltration, IFN-γ^+^/TNF-α^+^ CD8^+^ T-cells. Superior antitumor activity: strongest tumor inhibition across all models, including distant tumors and lung metastases. Excellent biocompatibility, with no significant body weight changes or organ toxicity. (↑ indicates an upregulation or increased expression/infiltration of the listed molecules or cell types as a result of treatment. For example, ER stress markers (IRE1α, ATF6, etc.) and immune components (DCs, CD4⁺, CD8⁺ T-cells) were elevated, reflecting activation of stress response and enhanced antitumor immunity.)	[164]
Mitoxantrone thermosensitive liposome (MTX-TSL)	Local hyperthermia at 41 °C, applied just before drug administration. External water-bath heating (localized heating).	BDF1 mice bearing RM-1 prostate tumors.	Hyperthermia-enhanced tumor accumulation of MTX-TSL. Significant tumor growth suppression in MTX-TSL + heat group vs. free drug ~80 % drug release within 30 min at 41 °C; ~96 % at 45 min.	[165]
EGFR-targeted PLGA nanoparticles encapsulating paclitaxel and perfluoropentane (PFP)	External ultrasound.	In vivo triple-negative breast cancer (TNBC) xenograft model in mice (MDA-MB-231).	PTX TNPs + ultrasound achieved the most potent tumor growth inhibition (tumor volume ~2.66 ± 1.72 vs. ≥5 in other groups). Marked reduction in microvessel density (CD31) and proliferation marker (Ki-67). Significant induction of apoptosis, with minimal systemic toxicity (no elevated ALT/AST or organ histopathology).	[166]
Thermosensitive hydrogel	Passive thermosensitive gelation (body heat triggered).	Orthotopic SKOV3-luc ovarian cancer model in athymic nude (BALB/c) mice via intraperitoneal inoculation.	Sustained release: Hydrogel erosion over 7 days with progressive siRNA accumulation in nodules. Tumor growth delay: Significant tumor suppression in treatment group (single injection), sustained through day 56 post-inoculation. Safety: No systemic toxicity or histopathological damage in major organs.	[163]
Janus micelles encapsulated within a thermosensitive hydrogel	Thermosensitive gelation: The micelle–hydrogel system transitions into a gel at physiological temperature (~37 °C).	Subcutaneous GBM model in mice.	Tumor volume reduction: 89.5 ± 3.34%. Immunostimulatory effects: Increased expression of CD80, NF-κB, IFN-γ, and TNF-α in both tumor and spleen tissues. Mechanism: Sequential release led to enhanced early apoptosis and reduced necrosis; dual-drug delivery amplified chemo-immunotherapy response.	[162]

**Table 8 ijms-26-07322-t008:** Summary of completed clinical studies of thermosensitive liposomal drug delivery for cancer therapy.

Nanocarrier Type	Formulation	Payload	Indication	Trial Phase	NCT	Ref.
Lysolipid-based thermosensitive liposome	ThermoDox^®^	DOX	Hepatocellular carcinoma (HCC)	Phase III	NCT02112656	[167]
Thermosensitive liposomes	ThermoDox^®^ (TARDOX)	DOX	Liver tumors	Phase I	NCT02181075	[33]
Thermosensitive liposomes	ThermoDox^®^	DOX	Recurrent chest wall breast cancer (CWR)	Phase I/II	NCT00826085	[168]
Lyso-thermosensitive liposome	ThermoDox^®^	DOX	Metastatic breast cancer	Phase I	NCT03749850	[169]

## Data Availability

Not applicable.

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
