# Peer review of "A Comprehensive Review of Smart Thermosensitive Nanocarriers for Precision Cancer Therapy"

_ijms, 2025, doi:10.3390/ijms26157322_

Round 1

Reviewer 1 Report

Comments and Suggestions for Authors

The manuscript author presents the latest advancements in thermosensitive nanocarriers for precision cancer therapy, with a particular focus on the promising applications of combining hyperthermia with temperature-responsive nano-delivery systems. The author provides a comprehensive comparison of various types of organic nanocarriers (e.g., micelles, liposomes, dendrimers, etc.), highlighting their advantages, limitations, and synthesis strategies. This work demonstrates high academic value and practical significance for guiding future research. However, there are some shortcomings.

1.The entire text needs to be concisely expressed.

2. Figure 1 and 2 have been redrawn and annotated.

3. Tables 1, 2, 4, 5 and 7 need to be redrawn.

4.Some English abbreviations were not defined for the first time, such as DDS, SDD, EPR, etc.

5.Rewrite the references in accordance with the format of this journal.

6. Improve the quality of the charts.

7.Lack of clinical application data. At present, the research progress mainly focuses on the laboratory level, and there is a lack of detailed discussion on the performance or obstacles of these nanosystems during the clinical trial stage.

8.The discussion can be further deepened, such as the insufficiently in-depth analysis of limitations, the challenges in large-scale production of nanocarriers, long-term biological safety issues, and the clinical difficulties in controlling the heat therapy dosage.

Comments on the Quality of English Language

/

Reviewer 2 Report

Comments and Suggestions for Authors

The proposed review covers a broad spectrum of smart nanocarriers, including polymer-based and lipid-based systems, highlighting their characteristics, advantages, and disadvantages. This provides a useful overview for researchers interested in nanomedicine for cancer therapy. The emphasis on temperature-responsive systems, especially in combination with hyperthermia, addresses a promising and innovative approach in targeted cancer treatment. This focus is timely given the need for more precise and less toxic therapies.

However, the review is too descriptive and could not be published in its current form.

Comments:

  1. According to the review title it should be focused on the development thermosensitive formulations for cancer treatment. However Tables 1 and 2 are devoted to just particulate formulations and liposomes, without accent on their thermosensitivity. These tables and corresponding text should be revised to give information on the thermoresponsive systems.
  2. The capture of Figure 2 is completely wrong. There is nothing about mechanism of stimuli-responsive formulations, but just schematic explanation of their effect. However, the mechanisms for thermosensitivity are missing. In my opinion they should be given in the very beginning of the paper to provide further clarity and classification of the thermoresponisve systems to be used in cancer treatment.
  3. The mechanism of thermosensitivity should be given and discussed in separate paragraph. Phase transition in polymers and lipids, prerequisite factors for such phase separation and possibility of such behaviour in the organism should be given and discussed.
  4. Figure 5. Why polymer chain at phase transition becomes a ring?
  5. The review lacks in-depth critical analysis of each nanocarrier type, such as specific challenges in clinical translation, regulatory hurdles, or detailed mechanisms of stimuli-responsiveness in each case.

Reviewer 3 Report

Comments and Suggestions for Authors

This review provides a timely and comprehensive overview of thermosensitive nanocarriers for targeted cancer therapy, emphasizing their synergy with hyperthermia. The topic is highly relevant to current research in precision oncology. The manuscript is well-structured, covers major nanocarrier classes in depth, and incorporates recent literature. While scientifically sound and potentially suitable for publication after revision, several areas require improvement to enhance clarity, depth, and impact.

  1. The review extensively describes how thermosensitive nanocarriers work but lacks critical discussion on why certain designs succeed/fail clinically. For example: 1) Why has only ThermoDox® (lysolipid TSL) advanced significantly in clinical trials despite decades of research on polymeric micelles/dendrimers? 2) Discuss key hurdles: Stability in blood (CMC issues for micelles), scale-up challenges (dendrimers), drug loading limitations (SLNs), and immunogenicity risks.
  2. Tables 4-8 primarily report in vitro release profiles/cell line studies. This limits the assessment of true therapeutic potential. I suggest to include a summary table comparing key in vivo efficacy parameters (tumor regression, survival, biodistribution) across nanocarrier types.
  3. The authors states a focus on "natural nanoparticles", however, the review predominantly covers synthetic systems (liposomes, PLGA, PNIPAM, PAMAM dendrimers). Natural polymers (chitosan, alginate, gelatin) appear minimally. I suggestion to include truly natural carriers (e.g., exosomes, protein-based NPs) and justify their advantages.
  4. Define all abbreviations at first use (e.g., UCST/LCST in Abstract, EPR in Introduction, TME in Fig 3 caption).
  5. Additional schematics or diagrams illustrating the mechanisms of hyperthermia and thermosensitive nanocarriers could enhance understanding.
  6. The authors need add a more in-depth discussion of the challenges and limitations associated with their clinical translation.

Reviewer 4 Report

Comments and Suggestions for Authors

In this manuscript, the authors have summarized the development of thermosensitive nanocarriers for cancer therapy. Generally, the manuscript is well written and informative. Before publication, only a few improvements are recommended. The details are as follows:

1. The translational prospects should be discussed to review the advantages and challenges for clinical application. The authors have mentioned ThermoDox; however, the details of ThermoDox in clinical trials need to be further introduced and discussed, highlighting the reasons behind the unsatisfactory outcomes. Additionally, more recent representative formulations should be included.

2. Thermosensitive formulations are usually combined with hyperthermia cancer therapy to achieve precise control over cargo release. Therefore, a more detailed introduction to hyperthermia cancer therapy should be added. In particular, clinical techniques such as focused ultrasound and electromagnetic ablation should be discussed.

Round 2

Reviewer 1 Report

Comments and Suggestions for Authors

The manuscript author presents the latest advancements in thermosensitive nanocarriers for precision cancer therapy, with a particular focus on the promising applications of combining hyperthermia with temperature-responsive nano-delivery systems. The author provides a comprehensive comparison of various types of organic nanocarriers (e.g., micelles, liposomes, dendrimers, etc.), highlighting their advantages, limitations, and synthesis strategies. This work demonstrates high academic value and practical significance for guiding future research.The author has made significant revisions to the proposed content. It is recommended that minor adjustments be made before it is published.

1.Check the format of the references.

Comments on the Quality of English Language

/

Author Response

Please find the answer to the reviewers' comments below

Reviewer 2 Report

Comments and Suggestions for Authors

Authors didn’t met my comments properly. The scientific explanation of the mechanism is missing. Table 1 doesn’t referring towards the topic of the paper. Moreover, the hydrogels in the table is a big mistake. May be authors meant nanogels? They also didn’t changed the figures. They trying to say that ring is a coil, but it is really looking as a ring, so why not to redraw it properly. In general the topic is interesting, but initial introduction into the topic and scientific explanation of the effects is very poor. This decrease the value of the review.                                                                              

Author Response

Please find the answer to the reviewers' comments below.
